# Partitioning the impacts of spatial and climatological rainfall variability in urban drainage modelling

Nadav Peleg[1], Frank Blumensaat[2,3], Peter Molnar[1], Simone Fatichi[1], and Paolo Burlando[1]

[1]ETH Zurich, Institute of Environmental Engineering, Hydrology and Water Resources Management, Zurich, Switzerland
[2]Swiss Federal Institute of Aquatic Science and Technology, Eawag, Dübendorf, Switzerland
[3]ETH Zurich, Institute of Environmental Engineering, Urban Water Systems, Zurich, Switzerland

*Correspondence to:* Nadav Peleg (nadav.peleg@sccer-soe.ethz.ch)

**Abstract.** The performance of urban drainage systems is typically examined using hydrological and hydrodynamic models where rainfall input is uniformly distributed, i.e. derived from a single or from very few rain–gauges. When models are fed with a single uniformly distributed rainfall realization, the response of the urban drainage system to the rainfall variability remains unexplored. The goal of this study was to understand how climate variability and spatial rainfall variability, jointly or individually considered, affect the response of a calibrated hydrodynamic urban drainage model. A stochastic spatially distributed rainfall generator (STREAP) was used to simulate many realizations of rainfall for a 30-year period, accounting for both climate variability and spatial rainfall variability. The generated rainfall ensemble was used as input into a calibrated hydrodynamic model (EPA SWMM) for simulating surface runoff and channel flow in a small urban catchment in the city of Lucerne, Switzerland. The variability of peak flows in response to rainfall of different return periods was evaluated at three different locations in the urban drainage network and partitioned among its sources. The main contribution to the total flow variability was found to originates from the natural climate variability (on average over 74%). In addition, the relative contribution of the spatial rainfall variability to the total flow variability was found to increase with longer return periods. This suggests that while the use of spatially distributed rainfall data can supply valuable information for sewer network design (typically based on rainfall with return periods from 5 to 15 years), there is a more pronounced relevance when conducting flood risk assessments for larger return periods. The results show the importance of using multiple distributed rainfall realizations in urban hydrology studies to capture the total flow variability in the response of the urban drainage systems to heavy rainfall events.

## 1 Introduction

Urban drainage systems are designed to ensure safe wastewater disposal (focus: dry weather) and adequate storm water handling (focus: wet weather). Whereas the variability of dry weather flows is rather low and well predictable, rain-induced flow dynamics scale over several orders of magnitude and require stochastic analysis due to the high rainfall variability. The latter is often addressed by summarizing the rainfall input in the form of Intensity–Duration–Frequency (IDF) curves (e.g. Guo, 2006; Yazdanfar and Sharma, 2015), which are essentially relating maxima of rainfall intensity for a given duration to their return period (Koutsoyiannis et al., 1998). For urban drainage system design, engineers choose return periods for which they expect

the urban drainage system to perform with a certain reliability (e.g. an acceptable number of failures such as overflows or flooding in a given return interval).

A common practice to evaluate the performance of urban drainage systems for different forcing situations is by using a model with a hydrological component to simulate the runoff at the urban catchment scale, and a hydrodynamic component to simulate the flow in the drainage system itself. Rainfall is defined as the most important input required by these models (Vaes et al., 2001). It is recommended to use high-resolution rainfall data in space and time as an input because of the short concentration time of urban drainage systems, and because it reduces flow prediction uncertainty. The required spatial and temporal resolution depends on the size of the urban catchment, characteristics of the drainage system, and local climate. A general recommendation is to use rainfall data in a resolution similar to (or higher than) that produced by a typical X–band weather radar system, i.e. minutes in time and sub-kilometer in space (see discussion by Berne et al., 2004; Bruni et al., 2015; Ochoa-Rodriguez et al., 2015; Wright et al., 2014b).

Rainfall input may be given by observations of rain–gauges and weather radar; however this constrains the analysis to storms observed in a limited time period. On the other hand, stochastic modeling of space-time rainfall fields allows a full exploration of the potential impacts of space-time variability in rainfall on the urban drainage system. The *spatial rainfall variability* is defined as the variability derived from having multiple (stochastic) spatially distributed rainfall fields for a given point in time. The temporal component, here referred to as *climate variability*, is defined as the variability derived from having multiple (natural) climate trajectories generating different distribution of storms and rainfall intensities in time. It is also known as "internal variability" or "stochastic climate variability" (Deser et al., 2012; Fatichi et al., 2016; Hawkins and Sutton, 2009). The consideration is that observed rainfall represents only one trajectory of a given climate and producing stochastic rainfall based on observed rainfall statistics results in many realizations, each one equally probable (see for example Peleg et al., 2015).

The use of stochastic rainfall generators that account for spatial rainfall variability and/or climate variability in urban hydrology applications is still rather new. Wright et al. (2014a) used stochastic storm transposition to synthesize long records of rainfall based on radar rainfall fields over the metropolitan area of Charlotte, North Carolina (USA), in order to estimate the discharge return periods for points inside the urban catchment. McRobie et al. (2013) extended the earlier Willems method (Willems, 2001) to generate spatially distributed Gaussian rainfall cells based on weather radar data for the Counters Creek catchment sewerage system in London (UK). Simoes et al. (2015) produced stochastic urban pluvial flood hazard maps for the Cranbrook urban catchment (UK) using the McRobie et al. (2013) rainfall generator. Gires et al. (e.g., 2012, 2013) used a multifractal model to generate space-time rainfall fields for the same storm but with different spatial structures, to study their effect on the simulated flow in conduits in the Cranbrook catchment. The most recent stochastic rainfall generators that are able to produce rainfall fields in a high spatial and temporal resolution and may be useful for urban applications are STREAP (Paschalis et al., 2013, 2014), HiReS-WG (Peleg and Morin, 2014) and STEPS (Foresti et al., 2016; Niemi et al., 2016).

The main objective of this paper is to investigate the relative contribution of the spatial versus climatic rainfall variability for flow peaks at different locations in the drainage network and for different return periods. We apply a new and advanced stochastic rainfall generator to simulate rainfall inputs for a small urban catchment in Lucerne (Switzerland) and simulate flow dynamics in the sewer system. This work demonstrates the potential of using stochastic rainfall generators for urban

applications and the benefits gained compared to other methods, such as bootstrapping rainfall events from a long rainfall series.

## 2  Case Study

The case study is an urban catchment located near the city center of Lucerne, Switzerland (Fig. 1). The catchment in total covers 77.0 $ha$, whereas 30.2 $ha$ are connected to the combined sewer network: 11.5 $ha$ of total area (5.3 $ha$ impervious area) are connected to location A and 30.2 $ha$ (13.6 $ha$) are connected to locations B and C. The catchment drains towards Lake Lucerne, with higher gradients at the upper part and moderate to low gradients in the lower part. The drainage system consists of separate and combined sewers (storm water and foul sewage share one pipe infrastructure) with a total network length of 11.2 $km$; hereinafter only combined sewers are considered. Both storm water and wastewater flows are solely driven by gravity. An overflow structure is built in the lower part of the catchment to alleviate network capacity excess during heavy rainfall (CSO location in Fig. 1). In this case, the carry-on flow towards the sewage treatment plant is hydraulically constrained (location B in Fig. 1), and excess water is spilled via a side-flow weir followed by a small retention tank (approximate 100 $m^3$) into Lake Lucerne (location C in Fig. 1).

The flow rate at the outlet of the combined sewer system (location B) was monitored for a period of 12 months from July 2014 to June 2015. In order to reduce measurement uncertainty, the water level and flow velocity was recorded using two different combi-sensors with different monitoring techniques (in-situ Doppler-ultrasound technique, ex-situ ultrasound-radar technique) in parallel. The recording interval was set to 1 $min$ and 15 $min$ for the Doppler-ultrasound sensor and the ultrasound-radar sensor, respectively.

## 3  Methods and Data

A stochastic space-time rainfall generator was used to simulate multiple realizations of 2-D rain fields for a 30-year period. The rainfall was generated for four distinct cases which were defined in order to explicitly account for the climate variability, spatial rainfall variability and total variability of the flow. The generated rainfall was used as an input into a hydrodynamic model. For each of the four cases, IDF curves were computed for the annual maxima of the mean areal rainfall and Flow–Duration–Frequency (FDF) curves were computed for annual flow maxima simulated at three different network elements, representing different aspects in the assessing of the performance of urban drainage systems. The total flow variability was partitioned into the part originating from climate variability and the additional contribution due to spatial rainfall variability. The methods are illustrated in Fig. 2.

## 3.1 Rainfall Data

Rainfall data originate from two sources: a rain–gauge located about 2 $km$ west to the case study catchment (Fig. 1) and a C–band weather radar composite. Both devices are operated by MeteoSwiss, the Swiss Federal Office of Meteorology and Climatology.

The tipping bucket rain–gauge records rainfall in 10 $min$ intervals with a precision of 0.1 $mm$. A 34-year record was used in this study, covering the period 1981–2014. High-resolution 10 $min$ rainfall intensities were benchmarked with hourly rainfall data (validated record provided by MeteoSwiss) and obvious deviations were corrected. The length of the observed record allows an adequate estimation of the statistical rainfall characteristics, especially regarding high rainfall intensities of short durations and with return periods up to 30 years. Climatological stationarity has been assumed for the observed record.

High-resolution radar rainfall data (2 $km$ and 5 $min$) for an 8-year period (2003–2010) were derived from a third-generation weather radar system of MeteoSwiss (Gabella et al., 2005; Germann et al., 2006). Radar grid cells were examined for substantial ground clutter or beam blockage and errors were excluded. This data was only used for the study of the rainfall structure over the catchment and not for the calculation of IDF curves, as the accuracy of rainfall intensity recorded by the weather radar (binned data) is not sufficient to address extremes. Extreme rainfall intensity for a 1 $km$ spatial resolution can be analyzed in Switzerland (e.g. Panziera et al., 2016) using the fourth-generation weather radar system (Germann et al., 2015) or the gridded CombiPrecip products (Sideris et al., 2014). However, a longer period of high-resolution rainfall from the latter mentioned products would be required in order to properly account for the climate variability discussed in this study.

## 3.2 Stochastic Rainfall Generator

Rainfall fields in a high spatial and temporal resolution were generated using the STREAP model (**S**pace-**T**ime **Re**alizations of **A**real **P**recipitation). STREAP was presented by Paschalis et al. (2013) and used to generate rainfall over a large rural catchment for flood investigations (Paschalis et al., 2014) and to analyze the variability of extreme rainfall intensity over radar-pixel scales (Peleg et al., 2016). It is composed of three hierarchical modules describing: (i) the storm arrival process; (ii) temporal evolution of the mean areal intensity and fraction of wet area during a storm; and (iii) evolution of the space–time structure of rainfall during a storm.

For this analysis, rainfall was generated with a spatial resolution of 100 $m$ x 100 $m$ for a domain size of 1.5 $km$ x 1.5 $km$ (see grid in Fig. 1) and a temporal resolution of 10 $min$. For urban drainage applications 10 $min$ can be considered a rather coarse temporal discretization, however we searched consistency with the observed rainfall record which is only available in 10 $min$ resolution. The spatial resolution was chosen to roughly match the discretization required for the urban sub-catchments, i.e. about two sub-catchments per $ha$ resulting in 158 individual sub-catchments within 77 $ha$.

### 3.2.1 STREAP Calibration

The calibration process of STREAP using weather radar products was discussed in detail by Paschalis et al. (2013). Some modifications were made to tailor STREAP to the specific case study presented here. Due to the short period of the weather

radar records (8 years), the storm arrival process (first module) was calibrated using the rain–gauge data (34 years). That allowed a better representation of the statistics of storm probability of occurrence and duration.

Changes were also applied to the second module. Originally, mean areal intensity and fraction of wet area during a storm are simulated as a bi-variate auto-correlated stochastic processes that also depend on storm duration. Here, due to the small extent
of the spatial domain, the wet area ratio was assumed to be equal to zero during intra-storm periods and assumed to be equal to one during storms, i.e. during storms all grid cells over the catchment are experiencing rainfall. The mean areal intensity is simulated using an AR(1) model which simulates a normalized quantile time series that is later inverted using a mixed-exponential function (Furrer and Katz, 2008; Smith and Schreiber, 1974), which parameters are computed using rain–gauge data.

No modifications were needed for the last module. However, some model parameters (e.g. rainfall coefficient of variation) could not be directly estimated from the weather radar data as the spatial resolution of the radar product ($2\ km$) is too coarse compared to the model resolution ($100\ m$). Therefore the required parameters were first estimated using the weather radar data for a coarse spatial resolution and then downscaled to higher resolution using power law functions (Fig. S1 in the supplementary material) as described in Peleg et al. (2016). In addition, no direct measurements are available to estimate the small-scale
rainfall spatial correlation structure for this region. The spatial structure was estimated using data from three dense rain-gauge networks (Moszkowicz, 2000; Müller and Haberlandt, 2016; Peleg et al., 2013), recording rainfall over small spatial distances (i.e. in the scale of $10^1$–$10^2\ m$) and temporal scales (i.e. 5–10 $min$). The data is presented in Fig. S2.

### 3.2.2   STREAP Evaluation

The evaluation of STREAP, its ability to reproduce the rainfall intensity over the domain (with emphasis on the high rainfall
intensity), and its performance with regard to the natural climate variability of the annual maxima in rainfall intensity, are discussed below.

The ability of STREAP to reproduce the rainfall intensity over the domain is shown using the inverse cumulative distribution function for the 10 $min$ mean areal rainfall intensity (Fig. 3). Up to the 0.95 quantile, STREAP consistently underestimates the rainfall intensity, but this underestimation is minor (maximum difference 0.052 $mm\ h^{-1}$) and is not expected to bias the flow
in the catchment. When we focus on the 0.95–1 quantile range which reflects the heavy rainfall intensity, and especially on the upper 0.99–1 quantile range, which represent the extremes, STREAP performs very well, with small differences between simulated and observed values (maximum difference 1.177 $mm\ h^{-1}$). The maximum difference and the maximum error calculated for the 0.9995–1 quantile range, which covers entirely annual maxima rainfall intensities that were observed, are as low as 1.072 $mm\ h^{-1}$ and 1.54 % (respectively). An example of STREAP ability to simulate spatially distributed annual
maxima rainfall intensity over the catchment is given in Fig. 4. The ability of STREAP to reproduce the natural climate variability in relation to the annual maxima rainfall intensity is discussed and presented in the Supplementary Material (Fig. S3).

## 3.3 Rainfall Cases Classification

Four rainfall cases were defined in order to account for climate variability and spatial rainfall variability and to allow the investigation of their effect on the urban drainage:

– Case 1: Consists of one time series of rainfall derived from the Lucerne rain–gauge (observed data, 34 years long). For this case, rainfall was not spatially distributed using STREAP but was uniformly distributed, i.e. same rainfall intensity was assigned to all sub-catchments for a given time step. In this case the rain–gauge time series represents also the mean areal rainfall over the catchment. This is a common and critical assumption in hydrological studies, where point rainfall is used to represent areal rainfall (Rodriguez-Iturbe and Mejia, 1974; Peleg et al., 2016; Sivapalan and Bloschl, 1998; Svensson and Jones, 2010).

– Case 2: consists of 30 realizations of the same time series (rain–gauge observations) that was used in case 1, but spatially distributed using STREAP. Cases 1 and 2 differ in the spatial configuration of the rainfall (uniformly distributed vs. spatially distributed) which will later allow to explicitly analyze how the spatial rainfall variability affects the flow.

– Case 3: consists of 30 realizations of 30 years generated by STREAP. For this case, STREAP was set to generate only the mean areal rainfall and to uniformly distribute it over the sub-catchments (similar to case 1). Comparing the urban drainage response to the rainfall given from cases 1 and 3 will allow us to account for the climate variability component directly, as case 3 represents 30 alternative and equiprobable trajectories of the rainfall series given in case 1.

– Case 4: consists of 900 realizations accounting for both the spatial rainfall variability and the climate variability. Each of the 30 realizations generated for case 3 were re-generated 30 times using STREAP. The forcing has a different spatial distribution of the rainfall over the sub-catchments for each re-generation. This allows computing urban drainage dynamics subjected to the total variability.

## 3.4 Hydrodynamic Model

Flow simulations were conducted using USEPA's Storm Water Management Model (EPA SWMM), a dynamic 1-D model coupling rainfall–runoff processes with hydrodynamic channel flow (Rossman, 2010). EPA SWMM was chosen as it represents a standard open-source application in urban drainage modeling (e.g. Hsu et al., 2000; Liong et al., 1995; Meierdiercks et al., 2010).

EPA SWMM is composed of two modules: the surface runoff (hydrological) and the in-sewer flow (hydraulic) model. The hydrological model calculates the direct runoff under consideration of initial precipitation losses (i.e. evaporation and wetting losses) and soil infiltration (here using the Horton method). The resulting surface runoff is then used as input for the hydraulic model to simulate the pipe flow using the 1-D Saint-Venant equations. The diffusive wave approximation and a routing step of 10 $s$ was applied for all simulations. Surface flooding is accounted for by allowing excess water to leave a manhole in case sewer capacity is exceeded. Due to the lack of detailed land use and surface topography data at meter scale it was found

inadequate to further define a manhole-specific "ponding area" allowing the water to spread at the surface around a manhole. Hence excess water leaving the manhole is routed into a virtual sink and does not re-enter the system even though sewer capacity is available again.

The sewer model application is based on infrastructure data from the municipality's cadaster database. The model has carefully been calibrated and validated (split-sample approach) using the above mentioned one-year flow data record. Flow dynamics can be adequately reproduced throughout the year despite the rather coarse 10 $min$ rainfall input data resolution. More details on the catchment, particularly on the urban land use characteristics, the monitoring set-up, model calibration procedure are given in Tokarczyk et al. (2015). The runoff-generating surfaces are distributed over the entire catchment. This is represented by 158 individual sub-catchment entities with an area ranging between 0.02 and 0.84 $ha$. The rainfall fields generated by STREAP were intersected with the sub-catchment areas and rainfall intensity was assigned for each sub-catchment based on the weighted sum of the intersect area (cf. Gires et al., 2012). EPA SWMM was set-up for a continuous long-term simulation of 30 and 34 years, respectively, depending on the examined rainfall case. Unlike for the design-storm approach or the isolated analysis of single storm events as researched in many previous studies, antecedent hydrological conditions in the catchment and the drainage network are implicitly taken into account to fully address potential climatological changes also regarding dry spells.

## 3.5 Computation of IDF and FDF Curves

The Generalized Extreme Value (GEV) distribution (Jenkinson, 1955) is commonly used in hydrological studies to model extreme rainfall intensity (e.g. Koutsoyiannis and Baloutsos, 2000; Marra and Morin, 2015; Marra et al., 2016) and flows (e.g Zaidman et al., 2003) since it covers the Gumbel, Fréchet and Weibull distributions (Katz et al., 2005). IDF and FDF curves were calculated by fitting a GEV distribution to the series of annual maxima of the mean areal rainfall intensity and the conduit flow time series (respectively). The fitting of the parametric distribution is a required step for the partition analysis to be conducted (see next section) as it results in a continuous estimates of the curves quantiles (i.e. the return period).

IDF curves were calculated for two datasets: observed data derived from the Lucerne rain–gauge and simulated data that were generated using STREAP. For the observed dataset, one IDF curve was computed for the 34 years of records. For the simulated dataset, 30 IDF curves were calculated for the 30 stochastic realizations (of 30 years each). The curves were calculated for a 10 $min$ duration.

FDF curves were calculated for the simulated flow at three locations which were chosen according to their function within the drainage network (see Fig. 1): (i) about 200 $m$ upstream of the combined sewer overflows (CSO) structure in a sewer section that was previously identified as prone to pipe surcharge (location A - inner network node); (ii) about 200 $m$ downstream of the CSO structure (location B – carry-on flow to sewage treatment works); and (iii) at the CSO outlet to lake (location C - overflow). The number of derived FDF curves follow the rainfall cases as described in Section 3.3, i.e. 1 for the first case (34 years), 30 for the second and third cases (30 years each) and 900 for the fourth case (30 years each). FDF curves were calculated for a 5 $min$ duration.

Note that no condition was imposed on the time concurrency of annual maxima of mean areal rainfall intensity and conduit flow, i.e. annual peak flow can precede, overlap or follow the annual maxima of mean areal rainfall intensity.

## 3.6 Variability Partitioning

The partition method used in this study follows the guidelines suggested by Fatichi et al. (2016). We assume that there are interactions between the two sources of variability, i.e. they cannot be treated independently, as the spatial pattern of the rainfall annual maxima is dependent on the extreme rainfall intensity that is driven by a given climate trajectory. An illustrative example for the partition method described in the following is given in Fig. 5.

The climate variability, $CLM$, is defined as the 5–95 quantile range of the flow that is calculated using the 30 spatial uniform climate realizations simulated for case 3 (i.e. the outcome is one flow range for a given return period). For each of the 30 climate realizations, the spatial flow variability, $SPT$, is defined as the 5–95 quantile difference of the flow calculated using the spatially variable rainfall simulated for case 4. The outcome is 30 different flow ranges, $SPT_{RP}^1, SPT_{RP}^2, ..., SPT_{RP}^{30}$ , one for each climate trajectory and for a given return period, $RP$. The 5–95 quantile range was used because of the different sample sizes between case 3 (30 realizations) and case 4 (900 realizations). The ratio between the climate variability, $CLM$ and the total variability, $TOT$, for each return period can then be estimated as:

$$\varphi_{CLM,RP} = \frac{CLM_{RP}}{TOT_{RP}} \tag{1}$$

where the total variability for a given return period is the difference between the maximum and minimum spatial variability simulated per return period from all climate trajectories:

$$TOT_{RP} = \max SPT_{RP} - \min SPT_{RP} \tag{2}$$

The total variability for a given return period, $TOT_{RP}$ , will be always smaller than the sum of the flow variability from case 4, because there is a dependency between the two sources of uncertainty. Note that $1 - \varphi_{CLM,RP}$ is representing the unique contribution of spatial variability to the total variability for a given return period, however the total spatial variability, $\frac{\sum_{i=1}^{N} SPT_{RP}^i}{N}$ , is larger or equal to $1 - \varphi_{CLM,RP}$ (see Fatichi et al., 2016).

## 4 Results and Discussion

In the following, we present computed IDF (rain) and FDF (flow) curves and discuss the contributions of individual rainfall variabilities to the modelled sewage flow variability at three different locations: A – inner network node (Fig. 7 and S4), B - carry-on flow (Fig. 6 and 8), and C - combined sewer overflow (Fig. 9 and S5). The partitioning of the flow variability is presented for all three locations (Fig. 10).

The effect of spatial rainfall variability on the flow can be directly estimated by examining the flow variability from case 2 (Fig. 6c, S4c and S5c). The effect of spatial rainfall variability is derived from the analysis of flow extremes occurring in a continuous time series of 30 years. The variability in annual flow maxima is computed from the spread in simulations for a

given return period. This variability is expressed as the difference between the highest and lowest flows simulated for the 30 realizations for a given return period. The effect of spatial rainfall variability on urban hydrology was researched in the past (e.g. Bruni et al., 2015; Gires et al., 2012; Simoes et al., 2015; Willems and Berlamont, 2002) leading to the conclusion that this variability should be taken into consideration when running urban hydrological models. Indeed, for return periods between

2 and 30 years, the peak flow variability was found to vary between 18.3 and 55.1 $l\ s^{-1}$ at location A (Fig. S4c) and between 91.2 and 179 $l\ s^{-1}$ at location C (Fig. S5c). At location B, peak flow variability was found to be lower (between 2.9 and 6.2 $l\ s^{-1}$, Fig. 6c) due to the fact that flow is hydraulically constrained by the upstream located throttle pipe.

The effect of the climate variability over the catchment is calculated from the 30 rainfall realizations stochastically simulated for cases 3 and 4 (left panel in Fig. 7–9). Similar to the flow variability, the climate variability is expressed as the difference

between the highest and lowest mean areal rainfall found for a given return period. In agreement with Peleg et al. (2016), the climate variability was found to increase with longer return periods, from 11.8 $mm\ h^{-1}$ for the two year return period to 47.2 $mm\ h^{-1}$ for the 30 year return period.

The individual effect of the climate variability on the flow is estimated from case 3 (Fig. 7b, 8b and 9b). For the return periods of 2 to 30 years, the flow variability at location C, resulting only from climate variability, was found to be in the range 278.9–

420.3 $l\ s^{-1}$. For most of the return periods this variability more than doubles the flow variability resulting from the spatial rainfall variability. The results for location C suggest that the role of climate variability is considerably more important than the role of spatial rainfall variability. The flow variability for the return periods 2–30 years for locations A and B were found to be in the range of 33.3–48.5 and 7.3–11.6 $l\ s^{-1}$ (respectively). As for location C, the flow variability resulting from climate variability is higher than the flow variability resulting from the spatial rainfall variability. However, the relative differences

in variability around the median peak flow, calculated for the 30 years return period, reveal that the differences between the individual variabilities are much less pronounced for locations A and B (1.7% to 3.2% and 0.8% to 2.1%, respectively) in comparison to location C (3.5% to 10.7%). These differences regarding the absolute flow variability are expected as location B is located downstream of a hydraulic constraint (throttle pipe at CSO structure), thus flow is eventually levelled out, while at location A runoff is drained directly from its contributing sub-catchment without any buffering but still constrained due to

surface flooding, i.e. excess flows leave the manhole through the lid and do not contribute to the actual peak flow.

The total flow variability is calculated using the data of case 4 (Fig. 7c, 8c and 9c). As expected, the total flow variability (e.g. location B: case 4, Fig. 8c) is larger than the flow variability resulting from either the spatial rainfall variability (case 2, Fig. 6c) or from the climate variability alone (case 3, Fig. 8b). The partitioning of the total flow variability into its components is presented for all three locations in Fig. 10. Results indicate that climate variability is the dominant contributor of the total

variability of flow in the catchment. This applies to peak flows analyzed at all three locations in the urban drainage system. The highest ratio between climate variability and total variability is for location B, 83% for 2.2 years return period, and decreases for longer return periods to 57% for the 30 years return period. This decreasing trend was found to be less prominent for locations A and C, but statistically significant for all three locations as supported by a trend analysis using the Mann-Kendall test (Kendall, 1975; Mann, 1945). For location A, the relative mean ratio between the climate and the total variability was

found to be around 81%. For location C, climate variability accounts for 75% for the 2 to 10 year return periods, decreasing to

62% for the 30 year return period. Averaged over all three locations and all return periods the mean ratio between the climate variability and the total variability is 74%, leaving 26% contribution due to the addition of spatial variability. The results of the partitioning suggest that using traditional methods to quantify variability in urban drainage, such as bootstrapping, will likely result in an underestimation of the variability (and uncertainty) as only the climate variability will be represented. This is especially important for return periods that are longer than 10 years. While the use of spatially distributed rainfall data can supply valuable information for sewer network design (based on rainfall with return periods from 5 to 15 years), it will become even more important when performing flood risk assessments of extreme events (larger return periods). A 30-year record was used in this study, which can be regarded as the minimum period for IDF/FDF analysis. Since uncertainties in climate statistics decrease with a longer observational record (e.g. Marra et al., 2016), the contribution of the additional spatial variability for larger return periods might be even greater than presented here. However, a longer period of observation is required to confirm this assertion.

The rainfall generator was used to simulate rainfall for the weather radar subpixel scale, i.e. in a finer spatial resolution than can be estimated using the MeteoSwiss radar. The rainfall data required for a complete validation of the rainfall generator for this resolution can be obtained from a dense rain–gauge network (for networks examples see Muthusamy et al., 2017; Peleg et al., 2013) but such a network is not available in the analyzed region. Four aspects are discussed in the light of missing information for the subpixel scale (i.e. rainfall downscaling process): (i) the rain fields are simulated following a lognormal distribution. We assume that the non-zero part of the subpixel spatial rainfall distribution follows the observed lognormal distribution that is recorded by the weather radar for this region (as in Paschalis et al., 2014; Peleg et al., 2016). A different spatial rainfall distribution will significantly affect the results of the extreme rainfall; (ii) we assume that occurrence and intensity statistics are equal for each of the grid cells, i.e. no spatial correlation is applied for the rainfall occurrence or intensity. This means that orography, distance from the lake, and urban micro-climate effects are not considered; (iii) we assume that the rainfall spatial correlation structure for this region follows the average structure obtained from estimates made in dense rain–gauge networks in Poland, Germany and Israel (Moszkowicz, 2000; Müller and Haberlandt, 2016; Peleg et al., 2013). The exact impact of the spatial correlation structure at the radar subpixel scale in urban drainage studies is yet to be determined; and (iv) we assume that the power-law used for the scaling of the rainfall coefficient of variation is continuous from the weather radar to its subpixel scale, and it is not affected by a scale-break. Overestimation of the rainfall coefficient of variation will affect the rainfall spatial variability and therefore impact the partitioning results.

No automatic calibration process exists for STREAP. The model requires not only high-resolution rainfall data but also an expert user for the calibration process, as modifications to the calibration procedure (e.g. scaling at higher spatial resolution) are needed in order to tailor STREAP to a given application.

The three locations analyzed in this study were deliberately chosen according to their functional hierarchy within the combined drainage system (i.e. inner network node, carry-on flow and overflow). By doing so, we can clearly differentiate the effect of spatial and climatological rainfall variability on elements depending on their function within the network. On the other hand, previous studies showed a tendency that conduits located upstream, not affected by hydraulically constraining structures, are more sensitive to rainfall spatial variability in comparison to conduits located downstream (e.g. Gires et al., 2012). While it

would be interesting to further investigate flow variability due to different spatial rainfall characteristics (e.g. the rainfall spatial correlation) at various upstream locations (similar as location A), this type of analysis would require larger drainage networks in comparison to the one presented here. Future studies will benefit from examining several different urban drainage systems with rainfall input from different high-resolution products to test the robustness of the findings.

Rainfall records were obtained from a rain–gauge that is located about 2 $km$ west of the case study catchment. It was chosen for three main reasons (i) its proximity to the catchment; (ii) it has a sufficiently long record (34-year) that is adequate for statistical climatology analysis; and (iii) records have been verified by MeteoSwiss ensuring sufficient consistency. In contrast to these advantages, the 10 $min$ temporal resolution of the rain data requires critical consideration when simulating the dynamics of the flow response (e.g. Ochoa-Rodriguez et al., 2015), particularly as the average flow response in the investigated

catchment is in the order of minutes. However, we achieve a reasonable hydraulic model performance when validating the model against flow observations at the catchment outlet (location B), considering peak flow, time-to-peak and flow balance (see Tokarczyk et al., 2015). Low flow volume errors (±5%) and Nash-Sutcliffe-Efficiencies of >0.8 for individual events, i.e. >0.7 for longer periods, support the fact that the flow dynamics are reproduced adequately. Remaining peak flow errors of up to 25% reflect existing deficiencies stemming from multiple sources, e.g. inadequate model structure, insufficient model calibration,

measurement errors in flow reference data and model input data uncertainty. Considering that the same hydrodynamic model has been used for all the simulations, it is likely that the error due to model structure and calibration do not introduce a consistent bias to the variability partitioning. A complete investigation of the model hydrodynamic uncertainties will provide additional insights but it will be difficult to constrain with the current length of available flow data

       The computational cost of running a rainfall generator combined with an urban drainage model may constrain the use of

the proposed approach for practical applications. But given the advances in the availability of computing capacity, also for non-scientific institutions, such application will become feasible in the near future. We have used a powerful 20 core desktop machine (Intel Xeon CPU E5-2687W) to run the 961 stochastic rainfall realizations with STREAP in approximately 4 days. We estimated that the time needed to run SWMM using the same stand-alone machine would have been about 4 months, which is impractically long duration. Therefore, we have used a high performance computing (HPC) cluster with hundreds of

computing nodes allowing SWMM simulations in less than 48 hours.

## 5   Conclusions

Output from a stochastic rainfall generator was used as input into an urban drainage model to investigate the effect of spatial rainfall variability and climate variability on peak flows in an urban drainage system located in central Switzerland. We found that the climate variability is the main contributor (74 % on average) to the total flow variability, but that the relative contribution

of the addition of spatial rainfall variability increases with return period. This implies that the use of spatially distributed rainfall data can supply valuable information for sewer network design (based on return periods of 5 to 15 years), but it will become even more relevant when assessing the risk of urban flooding as a consequence of intense rain events of larger return periods.

The analysis presented in this study focused on three different locations in the urban drainage system which reflect different system functions. Deviations in flow quantities and dynamics were expected and are, in fact, observed within the catchment depending on the corresponding location (i.e. up- or downstream of the overflow structure, or the overflow itself). Despite this, in agreement for all three locations we found that the climate variability is the dominant contributor to the flow variability for all return periods.

We present a single case study, a relatively small, but typical urban catchment located in the foothills of the Swiss Alps. We argue that the variability partitioning is likely to be similar for most small– to medium–sized urban catchments. That is to say, the climate variability will constitute the largest contribution to the overall flow variability also in other urban catchments, and spatial variability will gain more importance as longer return periods are being considered. Further investigations are needed to examine the contributions of the variability components in larger catchments (potentially more prone to spatial rainfall variability) with a more complex drainage network (potentially with more flow attenuation) and for different climates.

Stochastic spatially distributed rainfall generator should become an integral part of the urban hydrologist toolbox, particularly when estimating hazards of urban flooding. However, these models are still not commonly used by planning engineers for designing and evaluating urban drainage systems. We identify four main aspects that contribute to the reluctant acceptance in the field of urban drainage:

- High-resolution rainfall data are required (from a weather radar system or from a dense rain–gauge network) as well as an expert user for the calibration process. Setting up an automatic calibration process is unrealistic option due to the spatio-temporal differences between weather radar systems and the need to tailor the rainfall generator to specific locations.

- The high computational cost of running a rainfall generator combined with an urban drainage model may be prohibitive for common applications. Today the resources required for an efficient computation (e.g. HPC cluster) are often not available.

- The struggle to overcome old engineering paradigms towards accepting variability ranges as useful information for design and performance assessment.

- The difficulty of rainfall generators modelers to transparently convey the modeling chain, its results and uncertainties.

These aspects should be addressed in future applications of stochastic rainfall generators in order to make them more accessible to the urban drainage community.

*Acknowledgements.* This project is partly funded by the Swiss Competence Center for Energy Research – Supply of Electricity. We are grateful to MeteoSwiss, the Swiss Federal Office of Meteorology and Climatology, and the city of Lucerne for providing us with precipitation and infrastructure data. We furthermore would like to thank the Engineering Consultants from HOLINGER AG, Bern for assisting us with details on the hydraulic model and extracting operation data from the central data base. We thank the reviewers (Susana Ochoa-Rodriguez,

Li-Pen Wang and an anonymous reviewer) and to Marie-Claire ten Veldhuis, the editor, for their contributions leading to a significantly increased quality of the paper.

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

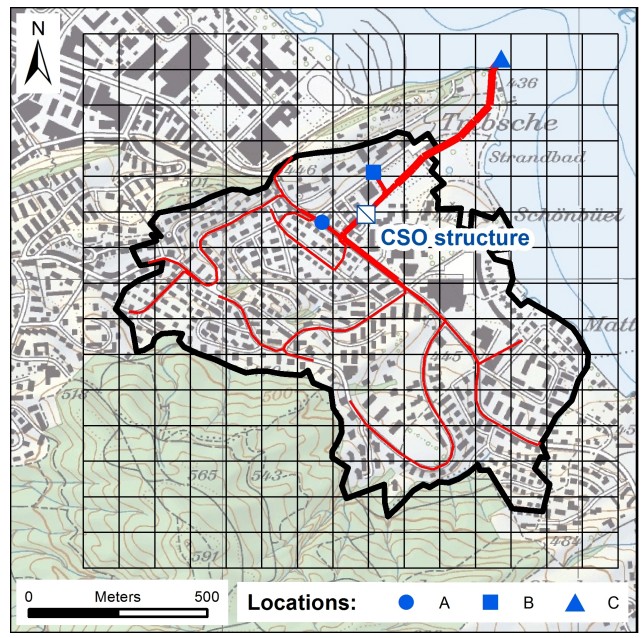

**Figure 1.** Location map of the case study catchment (bounded with black line). The black mesh represents the 1.5 x 1.5 $km^2$ domain (grid cell resolution of 100 x 100 $m^2$) for which stochastic rainfall was generated. The red lines represent the drainage system (thicker lines per pipe diameter) and the blue circle (inner network node), rectangle (carry-on flow) and triangle (combined sewer overflow) symbols represent the location for which the flow analysis was conducted. The combined sewer overflows (CSO structure, blue romb symbol) is located between locations A and B.

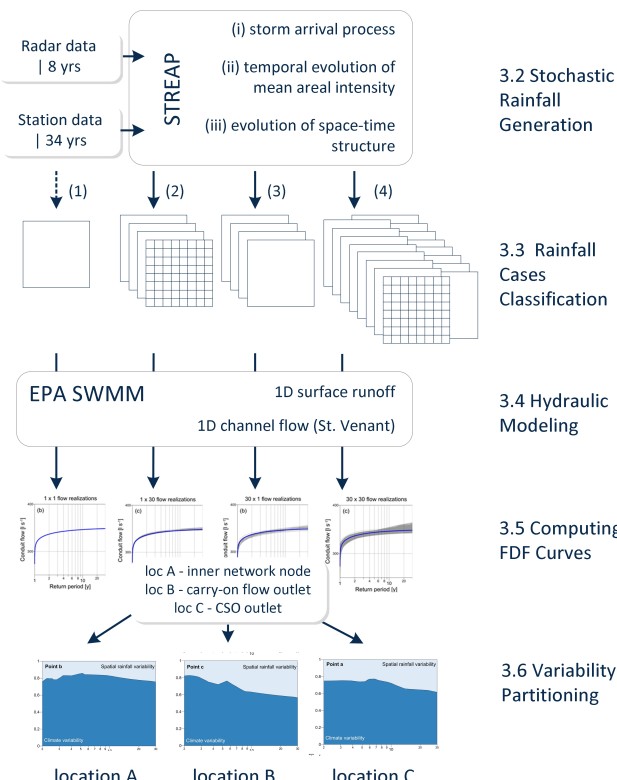

**Figure 2.** A schematic illustration of the methods used in this study: (i) STREAP model was used to simulate multiple realizations of 2-D rain fields based on radar and gauged data (Section 3.2); (ii) rainfall was generated for four distinct cases which were defined in order to explicitly account for the climate variability, spatial rainfall variability and total variability of the flow (3.3); (iii) EPA SWMM model was used to calculate the flow over the catchment (3.4); (iv) IDF and FDF curves were computed for the annual maxima of the mean areal rainfall and flow, respectively, at three different locations (3.5); and (v) the total flow variability was partitioned (3.6).

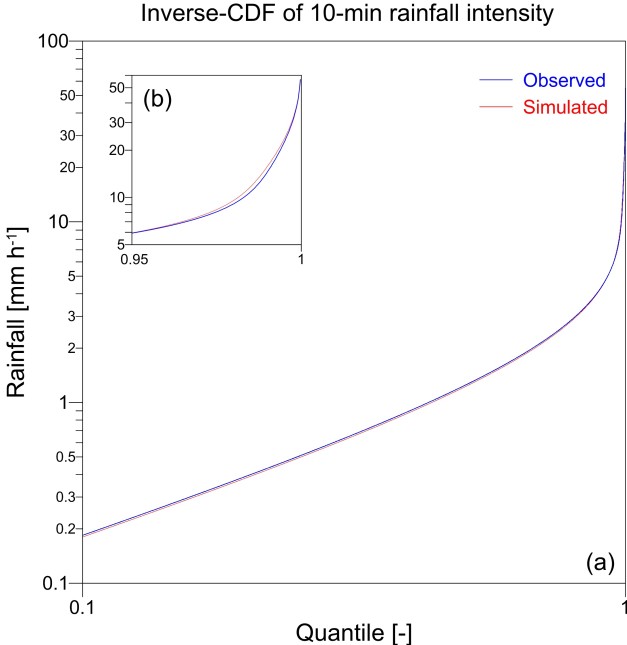

**Figure 3.** An inverse cumulative distribution function of the $10\ min$ mean areal rain intensity over the catchment [The 0.1–1 quantile is presented in (a) and the 0.95–1 quantile range is zoomed in (b)]. Blue line represents 34 years of observed data (1981–2014) and red line represents the median of 30 realizations of 30 years. The simulated mean 5–95 quantile range of the rainfall intensity of the 30 realizations is also presented (shaded red).

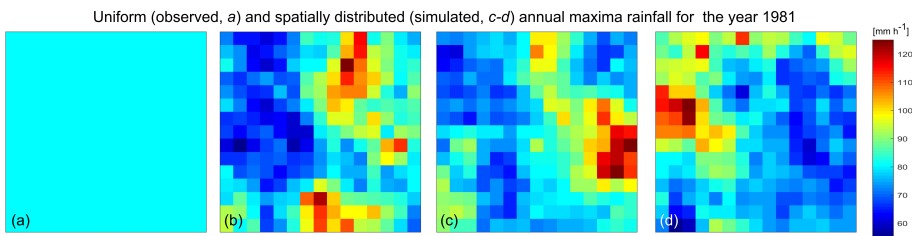

**Figure 4.** An example of STREAP ability to spatially distribute the annual maxima rainfall intensity over the catchment. The annual maxima recorded by Lucerne gauge for the year 1981 is $80.4\ mm\ h^{-1}$ for duration of $10\ min$. Without STREAP, this value is assumed to be uniformly distributed over the domain (panel a). STREAP accounts for the spatial distribution of rainfall, thus while the areal average is preserved for each time step, some grid cells (100 x 100 $m^2$ resolution) will record higher rainfall intensity and some lower values. Example of the footprint of the annual maxima rainfall intensity for three random realizations of the year 1981 generated by STREAP are presented in panels b–d.

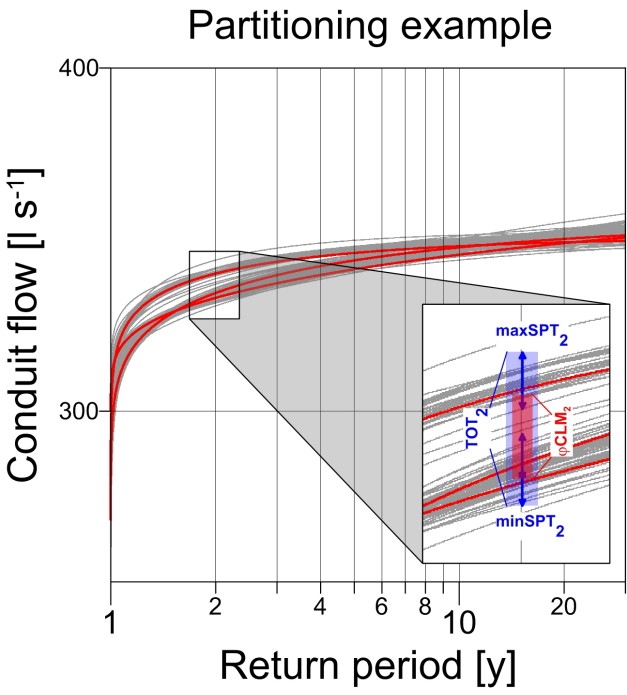

**Figure 5.** An example for the partition method (illustrative) for the 2 years return period (zoomed panel). 3 climate trajectories are plotted (red lines) for which the 5–95 quantile range is calculated ($\varphi_{CLM,2}$, red area). For each climate trajectory, 30 spatial realizations are plotted (grey lines). The 5–95 quantile range is then calculated for each of the 30 spatial realizations ($SPT_2^1, SPT_2^2, SPT_2^3$, plotted as blue arrows) and the total variability, $TOT_2$ (blue area), is defined by bounding the maximum and minimum flows defined by the spatial variability ($\max SPT_2$ and $\min SPT_2$, respectively). The partition of the climate variability, $\varphi_{CLM,2}$, out of the total variability is then calculated as a simple ratio between the two.

.

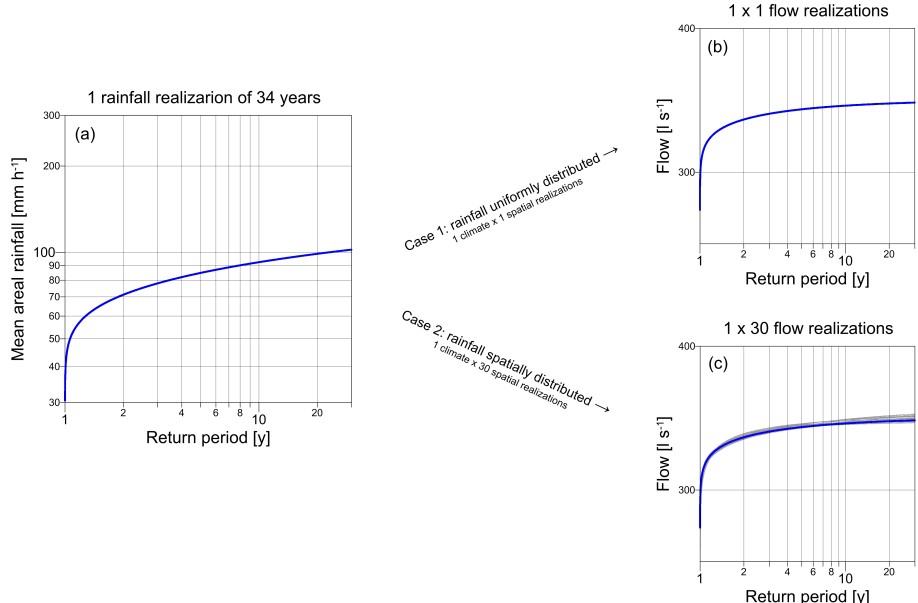

**Figure 6.** Rainfall and flow results for cases 1 and 2. In the left panel (a), the IDF curve computed for the mean areal rainfall over the catchment is presented. In the right panels, FDF curves for location B are presented for uniformly distributed rainfall (b) and spatially distributed rainfall (c). Blue line represents the IDF curve and the FDF curves computed from the observed uniformly distributed rainfall. Gray lines represent the FDF curves computed for the realizations with spatial rainfall variability.

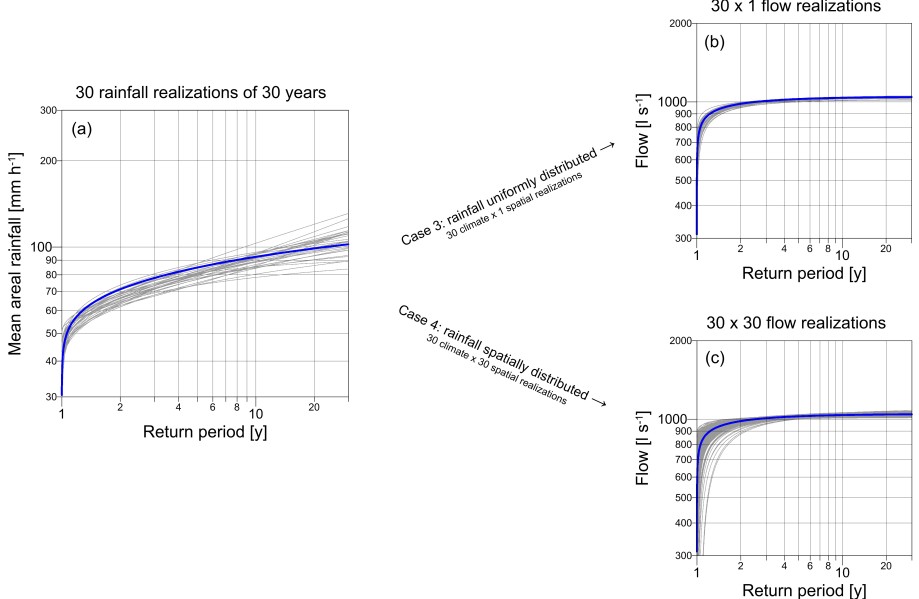

**Figure 7.** Same as Fig. 6, but for location A and cases 3 and 4.

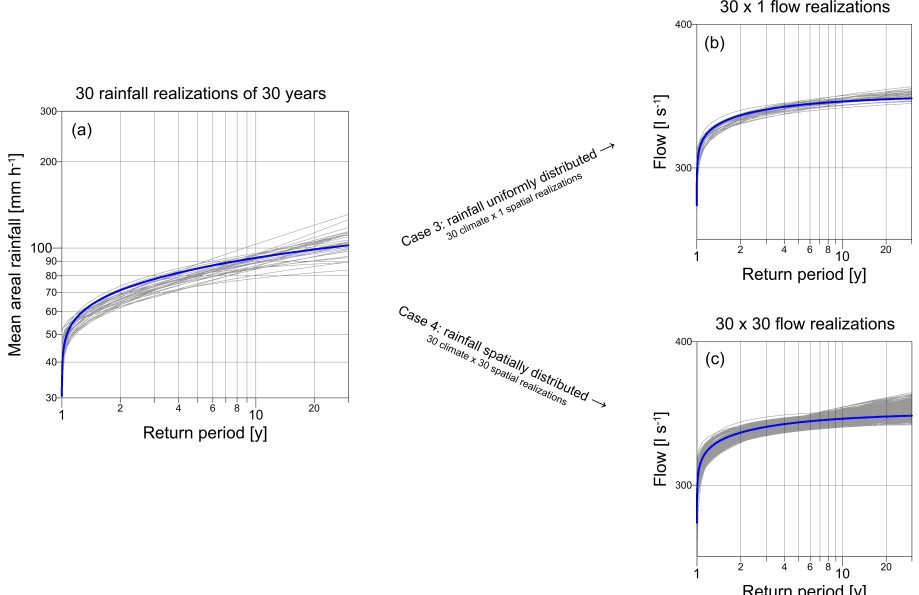

**Figure 8.** Same as Fig. 6, but for location B and cases 3 and 4.

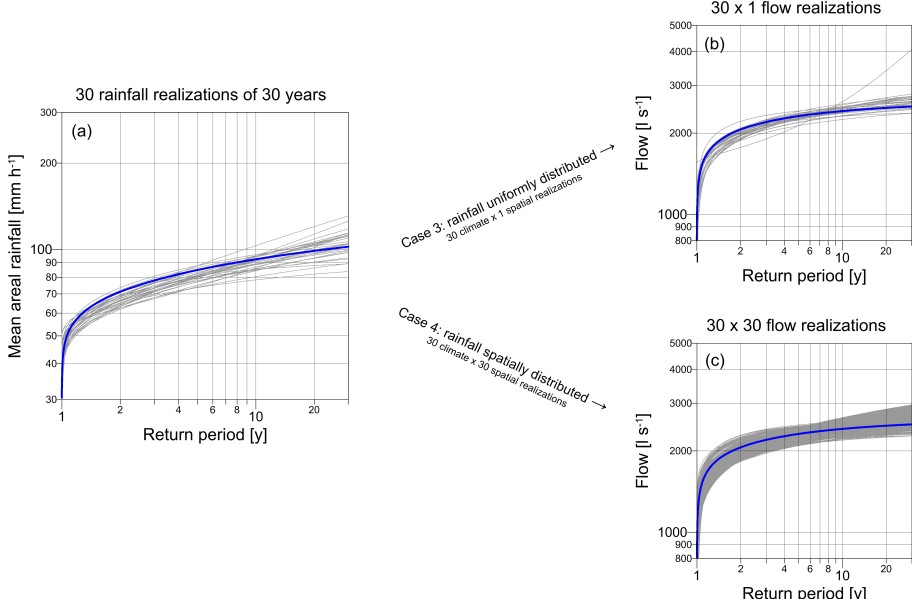

**Figure 9.** Same as Fig. 6, but for location C and cases 3 and 4. The poorly fitted GEV distribution for one realization presented in (b) was excluded from the flow variability partitioning analysis.

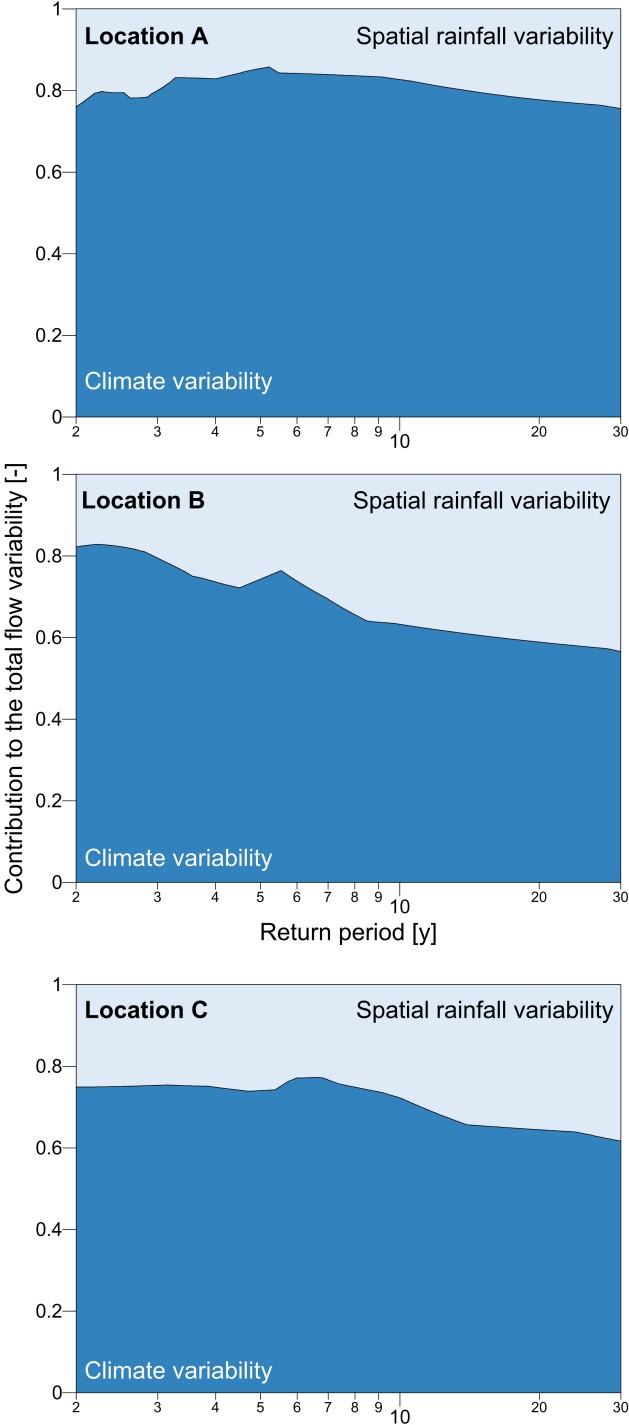

**Figure 10.** The ratio between the climate variability and the total flow variability for a given return period and for different locations within the urban drainage system is represented in dark blue. The remaining contribution is due to the addition of spatial rainfall variability (light blue).