# Peer review of "Partitioning the impacts of spatial and climatological rainfall variability in urban drainage modelling"

_Hydrology and Earth System Sciences, 2016_

## Referee Comment (RC1) · Anonymous Referee #1 · 4 Dec 2016

General Comments: The paper is very interesting and tries to add new knowledge to the field of urban hydrology. The use of stochastic rainfall generators and their impact in urban drainage is very important and recent. The authors try to quantify, not only the impact of spatial component of rainfall, but also its temporal component.

Specific comments: 1) It would be interesting to know the drainage area of each location where the flow analysis was conducted.

2) Why you didn't test more locations in the upstream part of the catchment. It would be interesting to see the climate and spatial contribution in smaller drainage areas (for example an upstream pipe and one not affected by hydraulic structures, such as CSOs.) This would be important, since some authors showed that upstream pipes are more sensitive to spatial variability (eg. Gires et al., 2012)

[Figure]

3) Is there flooding in any node? How did your swmm model deal with it? If there is flooding, what is the impact in the flow return period.

4) Figure 3 could be improved showing the inverse-CDF curve of all the 30 events, not only the mean.

5) In section 3 (1st paragraph) is not clear why do you use IDF and FDF curves, instead of the obtained/simulated values. I agree with the strategy, but a clear explanation should be added.

6) Figure 2 needs a better explanation

technical corrections:

1) Figure 2 needs more quality

2) In Figure 5 legend, where is "quantile range is than calculated for each" shoul be "quantile range is then calculated for each"

---

## Referee Comment (RC2) · S. Ochoa Rodriguez (Referee) · 13 Dec 2016

Review - Partitioning spatial and temporal rainfall variability in urban drainage modelling

by Susana Ochoa-Rodriguez & Li-Pen Wang

This study explores the influence of the spatial and natural climatological variability of rainfall on rainfall and associated flow return period estimation in urban areas. For this purpose, a stochastic rainfall generator was employed to generate rainfall time series with and without spatial variability. The resulting rainfall time series (corresponding to 4 different scenarios including combinations of spatial and climatological rainfall variability) were applied as input to the urban drainage model of a test catchment in Lucerne, Switzerland. Based on the results, both rainfall and flow return periods were computed and the individual influence of spatial and climatological rainfall variability on extremes

was quantified.

The study is very interesting and the results constitute a valuable contribution towards improved design of urban drainage systems. The paper is well written and we certainly enjoyed reading it.

I suggest that the authors clarify/address the following points prior to publication:

1. Please specify the drainage area of the points at which urban flows were analysed. As highlighted in previous studies (e.g. Berne et al. (2004); Gires et al. (2012); Wang et al. (2012); Ochoa-Rodriguez et al. (2015)), the drainage area of interest has a significant impact on the impact of spatial rainfall variability on simulated urban flows. In fact, in the figures provided in the supplement of the manuscript under consideration, it can be seen that the impact of spatial variability is somewhat different at the different locations at which flows are analysed. This is likely partly due to differences in the drainage areas associated to each point under consideration (this is, the areas upstream of the point of interest). Please provide information about the drainage areas under consideration and briefly analyse the impact of this factor on your results (a detailed analysis of this could also be suggested as 'future work').

2. While 2 km radar data were employed to calibrate the rainfall generator, 100 m spatial data were then generated. Please discuss the implications of this and whether the downscaling model that was employed accounts for scaling, thus making it appropriate to downscale down to 100 m (although the model was only calibrated based on 2km data).

3. The temporal resolution adopted in the study (10 min) may be too coarse for urban applications and may result in smoothing of urban flows, which may in turn result in underestimation of flow extremes (as indicated in Ochoa-Rodriguez et al. (2015), temporal resolutions <= 5 min are required for urban hydrological applications, with resolutions of 10 min leading to large underestimation of peak flows. Likewise, Wang et al. (2015) showed results of flow simulations resulting from rainfall inputs at tempo-

ral resolutions from 1 to 10 minutes and compared them against flow observations; the results associated to 10 min rainfall inputs largely underestimated observed flow peaks and resulted in 'distorted' hydrographs). I understand that the temporal resolution of choice was likely constrained by the resolution at which rain gauge rainfall records were available. Please discuss the implications and limitations of the temporal resolution of choice and clearly mention in the future work section that tests should be conducted at finer temporal resolutions.

4. The current title of the paper is rather misleading and I would suggest changing it to better reflect the purpose and focus of the study. For example, the focus on extreme values / return period should somehow be mentioned. Furthermore, I would suggest changing 'temporal variability' to 'natural climatological variability'. The term temporal variability conveys the idea of temporal resolution, which, as described above, is not the purpose of this study and is in fact one of its shortcomings.

REFERENCES:

Berne, A., Delrieu, G., Creutin, J.-D. & Obled, C. (2004). Temporal and spatial resolution of rainfall measurements required for urban hydrology. Journal of Hydrology, 299, 166-179.

Gires, A., Onof, C., Maksimović, Č., Schertzer, D., Tchiguirinskaia, I. & Simoes, N. (2012). Quantifying the impact of small scale unmeasured rainfall variability on urban runoff through multifractal downscaling: A case study. Journal of Hydrology, 442, 117-128.

Ochoa-Rodriguez, S., Wang, L.-P., Gires, A., Pina, R. D., Reinoso-Rondinel, R., Bruni, G., Ichiba, A., Gaitan, S., Cristiano, E., Assel, J. v., Kroll, S., Murlà-Tuyls, D., Tisserand, B., Schertzer, D., Tchiguirinskaia, I., Onof, C., Willems, P. & ten Veldhuis, M.-C. (2015). Impact of spatial and temporal resolution of rainfall inputs on urban hydrodynamic modelling outputs: A multi-catchment investigation. Journal of Hydrology, 531, 389-407.

Wang, L.-P., Ochoa-Rodríguez, S., Van Assel, J., Pina, R. D., Pessemier, M., Kroll, S., Willems, P. & Onof, C. (2015). Enhancement of radar rainfall estimates for urban hydrology through optical flow temporal interpolation and Bayesian gauge-based adjustment. Journal of Hydrology, 531, 408-426.

Wang, L.-P., Onof, C., Ochoa-Rodríguez, S., Simoes, N. E. & Maksimović, Č. (2012). On the propagation of rainfall bias and spatial variability through urban pluvial flood modelling. In 9th International Workshop on Precipitation in Urban Areas: Urban challenges in rainfall analysis, Saint Moritz, Switzerland.

——————————————————————

---

## Author Comment (AC1) · 21 Dec 2016

General Comments: The paper is very interesting and tries to add new knowledge to the field of urban hydrology. The use of stochastic rainfall generators and their impact in urban drainage is very important and recent. The authors try to quantify, not only the impact of spatial component of rainfall, but also its temporal component.

[reply] We thank the reviewer for his time and effort reviewing our manuscript.

Specific comments:

1) It would be interesting to know the drainage area of each location where the flow analysis was conducted.

[reply] Contributing area: 11.5 ha total area (5.3 ha impervious area) is connected to location A and 30.2 ha (13.6 ha) is connected to locations B and C, whereas the two latter locations are constrained through the overflow weir structure. We will add this information in manuscript.

2) Why you didn't test more locations in the upstream part of the catchment. It would be interesting to see the climate and spatial contribution in smaller drainage areas (for example an upstream pipe and one not affected by hydraulic structures, such as CSOs.) This would be important, since some authors showed that upstream pipes are more sensitive to spatial variability (eg. Gires et al., 2012)

[reply] Location A is located upstream of the CSO and is not affected by it. The goal in this study was to compare the spatial vs. climatic rainfall variability contribution to the total flow variability, thus we make do with one upstream location for the analysis. Examining the effects of contributing area on peak flows depending on the topological location within the network indeed requires analyzing further locations along the drainage system. This maybe be interesting - as shown in previous studies - however it is not a key aspect of this paper. We will address the reviewer's comment in the revised version of the manuscript.

3) Is there flooding in any node? How did your SWMM model deal with it? If there is flooding, what is the impact in the flow return period.

[reply] Node flooding (overflow and re-intake into the sewer system) can be simulated in SWMM, but it was not enabled in this study (ponded area was set to zero due to lack of adequate topography information). The flow return period is computed for the entire response of the drainage system, including the floods if occur at different nodes. A note will be added to the manuscript explaining how we dealt with node flooding throughout our SWMM simulations.

4) Figure 3 could be improved showing the inverse-CDF curve of all the 30 events, not only the mean.

[reply] It is represented (shaded red area) for the 5-95 quantile of all 30 events. The range is very thin and almost unnoticeable.

5) In section 3 (1st paragraph) is not clear why do you use IDF and FDF curves, instead of the obtained/simulated values. I agree with the strategy, but a clear explanation should be added.

[reply] A sentence will be added to rationalize the use in IDF and FDF curves.

6) Figure 2 needs a better explanation

[reply] Further explanation will be added to the figure caption.

Technical corrections:

1) Figure 2 needs more quality

2) In Figure 5 legend, where is "quantile range is than calculated for each" should be "quantile range is then calculated for each"

[reply] We will address the technical corrections in the revised manuscript.

---

## Author Comment (AC2) · 21 Dec 2016

This study explores the influence of the spatial and natural climatological variability of rainfall on rainfall and associated flow return period estimation in urban areas. For this purpose, a stochastic rainfall generator was employed to generate rainfall time series with and without spatial variability. The resulting rainfall time series (corresponding to 4 different scenarios including combinations of spatial and climatological rainfall variability) were applied as input to the urban drainage model of a test catchment in Lucerne, Switzerland. Based on the results, both rainfall and flow return periods were computed and the individual influence of spatial and climatological rainfall variability on extremes was quantified. The study is very interesting and the results constitute a valuable contribution towards improved design of urban drainage systems. The paper is well written and we certainly enjoyed reading it.

[reply] We thank the reviewers for their kind words and for the time and effort reviewing our manuscript.

I suggest that the authors clarify/address the following points prior to publication:

1. Please specify the drainage area of the points at which urban flows were analysed. As highlighted in previous studies, the drainage area of interest has a significant impact on the impact of spatial rainfall variability on simulated urban flows. In fact, in the figures provided in the supplement of the manuscript under consideration, it can be seen that the impact of spatial variability is somewhat different at the different locations at which flows are analysed. This is likely partly due to differences in the drainage areas associated to each point under consideration (this is, the areas upstream of the point of interest). Please provide information about the drainage areas under consideration and briefly analyse the impact of this factor on your results (a detailed analysis of this could also be suggested as 'future work').

[reply] The drainage area is: 11.5 ha total area (5.3 ha impervious area) are connected to location A and 30.2 ha (13.6 ha) are connected to locations B and C, whereas both locations are constrained through the overflow weir structure. A detailed analysis as suggested by the reviewers will require a larger catchment than the one studied here (might be suggested as a 'future work'). We will add this information in manuscript.

2. While 2 km radar data were employed to calibrate the rainfall generator, 100 m spatial data were then generated. Please discuss the implications of this and whether the downscaling model that was employed accounts for scaling, thus making it appropriate to downscale down to 100 m (although the model was only calibrated based on 2km data).

[reply] The rainfall model was scaled to 100 m resolution as mentioned in Section 3.2.1. We will add a figure in the supplementary material with an example of the model scaling (e.g. for the rainfall coefficient of variation) and will add a short explanation in the text for the choice of rainfall spatial correlation structure used.

3. The temporal resolution adopted in the study (10 min) may be too coarse for urban applications and may result in smoothing of urban flows, which may in turn result in underestimation of flow extremes (as indicated in Ochoa-Rodriguez et al. (2015), temporal resolutions <= 5 min are required for urban hydrological applications, with resolutions of 10 min leading to large

underestimation of peak flows. Likewise, Wang et al. (2015) showed results of flow simulations resulting from rainfall inputs at temporal resolutions from 1 to 10 minutes and compared them against flow observations; the results associated to 10 min rainfall inputs largely underestimated observed flow peaks and resulted in 'distorted' hydrographs). I understand that the temporal resolution of choice was likely constrained by the resolution at which rain gauge rainfall records were available. Please discuss the implications and limitations of the temporal resolution of choice and clearly mention in the future work section that tests should be conducted at finer temporal resolutions.

[reply] The rainfall temporal resolution was indeed set to 10 min following the temporal resolution of the records. We do agree that temporal resolution of rainfall data used as model input significantly influences flow dynamics including peak flows. We will add a short discussion of the implications and limitations of the 10 min temporal resolution used in this study.

4. The current title of the paper is rather misleading and I would suggest changing it to better reflect the purpose and focus of the study. For example, the focus on extreme values / return period should somehow be mentioned. Furthermore, I would suggest changing 'temporal variability' to 'natural climatological variability'. The term temporal variability conveys the idea of temporal resolution, which, as described above, is not the purpose of this study and is in fact one of its shortcomings.

[reply] We agree with the reviewer comment. A more suitable title for the manuscript will be consider and the term "temporal variability" will be replaced with "climatic variability" (or similar term).

---

## Author Response (AR1)

January 27, 2017

Dr Marie-Claire ten Veldhuis
Editor
Hydrology and Earth System Sciences (HESS)

RE: hess-2016-530

Dear Editor,

Please find enclosed the revised manuscript: "Partitioning the impacts of spatial and climatological rainfall variability in urban drainage modelling", previously entitled "Partitioning spatial and temporal rainfall variability in urban drainage modelling" by Nadav Peleg, Frank Blumensaat, Peter Molnar, Simone Fatichi and Paolo Burlando. The manuscript has been revised according to the comments of the two reviewers. We would like to thank the Editor and the two reviewers (Susana Ochoa Rodriguez and an anonymous reviewer) for their efforts and constructive comments. Detailed answers to the reviewers' comments are given below.

**Reviewer #1:**

General Comments: The paper is very interesting and tries to add new knowledge to the field of urban hydrology. The use of stochastic rainfall generators and their impact in urban drainage is very important and recent. The authors try to quantify, not only the impact of spatial component of rainfall, but also its temporal component.
We thank the reviewer for his time and effort reviewing our manuscript.
Specific comments:

1) It would be interesting to know the drainage area of each location where the flow analysis was conducted.
Contributing area: 11.5 ha total area (5.3 ha impervious area) is connected to location A and 30.2 ha (13.6 ha) is connected to locations B and C, whereas the two latter locations are constrained through the overflow weir structure. This information was added in manuscript.

2) Why you didn't test more locations in the upstream part of the catchment. It would be interesting to see the climate and spatial contribution in smaller drainage areas (for example an upstream pipe and one not affected by hydraulic structures, such as CSOs.) This would be important, since some authors showed that upstream pipes are more sensitive to spatial variability (eg. Gires et al., 2012)
Location A is located upstream of the CSO, but clearly not affected by water backing up due to the hydraulic constraint at the CSO structure. Examining the effects of contributing area on peak flows depending on the topological location within the network indeed requires analyzing further locations, and it is an interesting point to examine. However, we did not include this aspect into our analysis for two reasons: (i) we did not want to repeat analyses that had been done already in previous studies (e.g. by Gires et al., 2012); and (ii) the examined system is too small to obtain a representative answer on this aspect. Instead, we focus on the system response for three distinctively chosen, characteristic locations, differing regarding their functional hierarchy (A - upstream in the catchment, B – downstream of overflow structure at the overflow outlet, C – downstream overflow structure in the carry-on flow). By doing so, we believe that the analysis has an added value since effects of spatial and climatological rainfall variability can be differentiated depending on the *function* of the network element.
We still think it would be interesting to see if results shown by Gires et al. (2012) on the spatial variability in the upper part of the catchment, similar like location A, can be reproduced. However, we clearly think a larger urban drainage network, i.e. a larger number of nodes, should be examined following the same methodology to come to a representative conclusion with this regard. Finally, we addressed this issue in the revised version of the manuscript by discussing potential options to conduct this extended analysis depending on the topological location in the network:

"The three locations analyzed in this study were deliberately chosen according to their functional hierarchy within the combined drainage system (i.e. inner network node, carry-on flow and overflow). By doing so, we can clearly differentiate the effect of spatial and climatological rainfall variability on elements depending on their function within the network. On the other hand, previous studies showed a tendency that conduits located upstream, not affected by hydraulically constraining structures, are more sensitive to rainfall spatial variability in comparison to conduits located downstream (e.g. Gires et al., 2012). While it would be interesting to further investigate flow variability due to different spatial rainfall characteristics (e.g. the rainfall spatial correlation) at various upstream locations (similar as location A), this type of analysis would require larger drainage networks in comparison to the one presented here. Future studies will benefit from examining several different urban drainage systems with rainfall input from different high-resolution products to test the robustness of the findings".

3) Is there flooding in any node? How did your SWMM model deal with it? If there is flooding, what is the impact in the flow return period.

Yes, flooding occurs at several nodes when simulating the 30 year period(s). For example at location A, for the reference period 1980 – 2010 surface flooding was estimated for 34 events (0.1 – 1.5 hours of duration) using measured single gauge rainfall data. The number of nodes at which flooding is observed is limited; location A is one of the few locations (8 – 10 in total) where flooding occurs. Surface flooding is accounted for by allowing excess water to leave a manhole in case sewer capacity is exceeded. Due to the lack of detailed land use and surface topography data it was found inadequate to define a node-specific 'ponding area' (i.e. surface area that is available allowing the water to spread at the surface around a manhole). Hence surface flooding is taken into account but excess water leaving the manhole is routed into a virtual sink and does not re-enter through the manhole lid after sewer capacity is available again. We evaluated simulation runs with different simulation options to consider flooded water volume re-entering the system (namely: a) ponding area left undefined; b) definition of generally 100 m² 'ponding area') specifically focusing on deviations of flow rates at study relevant locations. Results from benchmarking simulations (see collection of figures below; notated with *F#*) clearly show that choosing different simulation options does not significantly affect the in-sewer flow rates, which had been selected as evaluation criterion in this study. We verified this for short periods, i.e. a series of events (example in F1 to F3, upper graphs), as well as for long terms (example in F1 to F3, lower graphs), whereas series labelled as '*ref data*' stand for simulations without having a ponding area defined and series labelled with '*sim data*' represent simulation for which a ponding area of 100m² had been defined. Performance metrics (Bias, Nash-Sutcliff-Efficiency, Index of Agreement, volume balance) indicate the low order of magnitude for flow rate, i.e. flow return period deviations. Still, for simulations for which excess water is not allowed to re-enter the system ('*ref data*', '*Vref*') the flow volume is – as expected - slightly lower. The following paragraph was added to the manuscript in Section 3.4 explain how surface flooding is considered in the study: "Surface flooding is accounted for by allowing excess water to leave a manhole in case sewer capacity is exceeded. Due to the lack of detailed land use and surface topography data at meter scale it was found inadequate to further define a manhole-specific "ponding area" allowing the water to spread at the surface around a manhole. Hence excess water leaving the manhole is routed into a virtual sink and does not re-enter the system even though sewer capacity is available again.".

[Figure]

F1: Location A        F2: Location B        F3: Location C

4) Figure 3 could be improved showing the inverse-CDF curve of all the 30 events, not only the mean.
It is represented (shaded red area) for the 5-95 quantile of all 30 events. The range is very thin and almost unnoticeable.

5) In section 3 (1st paragraph) is not clear why do you use IDF and FDF curves, instead of the obtained/simulated values. I agree with the strategy, but a clear explanation should be added.
A sentence rationalizing the use of a parametric distribution fitting to compute the IDF and FDF curves was added to Section 3.5: "The fitting of the parametric distribution is a required step for the partition analysis to be conducted (see next section) as it results in a continuous estimates of the curves quantiles (i.e. the return period)".

6) Figure 2 needs a better explanation
The caption of Figure 2 was changed as follows: "A schematic illustration of the methods used in this study: (i) STREAP model was used to simulate multiple realizations of 2-D rain fields based on radar and gauged data (Section 3.2); (ii) rainfall was generated for four distinct cases which were defined in order to explicitly account for the climate variability, spatial rainfall variability and total variability of the flow (3.3); (iii) EPA SWMM model was used to calculate the flow over the catchment (3.4); (iv) IDF and FDF curves were computed for the annual maxima of the mean areal rainfall and flow, respectively, at three different locations (3.5); and (v) the total flow variability was partitioned (3.6)".

Technical corrections:
1) Figure 2 needs more quality
The quality of Figure 2 was enhanced to 600 dpi.

2) In Figure 5 legend, where is "quantile range is than calculated for each" should be "quantile range is then calculated for each"
Thank you for pointing out this typo. The sentence was corrected.

**Reviewer #2 (Susana Ochoa Rodriguez):**

This study explores the influence of the spatial and natural climatological variability of rainfall on rainfall and associated flow return period estimation in urban areas. For this purpose, a stochastic rainfall generator was employed to generate rainfall time series with and without spatial variability. The resulting rainfall time series (corresponding to 4 different scenarios including combinations of spatial and climatological rainfall variability) were applied as input to the urban drainage model of a test catchment in Lucerne, Switzerland. Based on the results, both rainfall and flow return periods were computed and the individual influence of spatial and climatological rainfall variability on extremes was quantified. The study is very interesting and the results constitute a valuable contribution towards improved design of urban drainage systems. The paper is well written and we certainly enjoyed reading it.
We thank the reviewers for their kind words and for the time and effort reviewing our manuscript.

I suggest that the authors clarify/address the following points prior to publication:

1. Please specify the drainage area of the points at which urban flows were analysed. As highlighted in previous studies, the drainage area of interest has a significant impact on the impact of spatial rainfall variability on simulated urban flows. In fact, in the figures provided in the supplement of the manuscript under consideration, it can be seen that the impact of spatial variability is somewhat different at the different locations at which flows are analysed. This is likely partly due to differences in the drainage areas associated to each point under consideration (this is, the areas upstream of the point of interest). Please provide information about the drainage areas under consideration and briefly analyse the impact of this factor on your results (a detailed analysis of this could also be suggested as 'future work').
The drainage area connected to locations of interest is: 11.5 ha total area (5.3 ha impervious area) are connected to location A and 30.2 ha (13.6 ha) are connected to locations B and C, whereas locations B and C are constrained through the overflow weir structure. This information was added in the manuscript. Location A is located upstream of the CSO and is not affected by it. In the study we focus on the system response for three distinctively chosen, characteristic locations, differing regarding their functional hierarchy (A - upstream in the catchment, B – downstream of overflow structure at the overflow outlet, C – downstream overflow structure in the carry-on flow). By doing so, we can show that effects due spatial and climatological rainfall variability need to be differentiated depending on the *function* of the corresponding network element and not only depending on the contributing area. Examining the effects of contributing area on peak flows depending on the topological location within the network indeed requires analyzing further locations within the drainage system upstream of the hydraulically constraining structures. Although this analysis is interesting, it was not a key aspect of this paper. We think it will require a larger drainage network than examined in the presented case study. We addressed this issue in the discussion section in the revised version of the manuscript as follows:
"The three locations analyzed in this study were deliberately chosen according to their functional hierarchy within the combined drainage system (i.e. inner network node, carry-on flow and overflow). By doing so, we can clearly differentiate the effect of spatial and climatological rainfall variability on elements depending on their function within the network. On the other hand, previous studies showed a tendency that conduits located upstream, not affected by hydraulically constraining structures, are more sensitive to rainfall spatial variability in comparison to conduits located downstream (e.g. Gires et al., 2012). While it would be interesting to further investigate flow variability due to different spatial rainfall characteristics (e.g. the rainfall spatial correlation) at various upstream locations (similar as location A), this type of analysis would require larger drainage networks in comparison to the one presented here. Future studies will benefit from examining several different urban drainage systems with rainfall input from different high-resolution products to test the robustness of the findings".

2. While 2 km radar data were employed to calibrate the rainfall generator, 100 m spatial data were then generated. Please discuss the implications of this and whether the downscaling model that was employed accounts for scaling, thus making it appropriate to downscale down to 100 m (although the model was only calibrated based on 2km data).

The rainfall model was downscaled to 100 m resolution using power laws functions, as mentioned in Section 3.2.1. A figure was added in the supplementary material (new Figure S1) showing the scaling of the rainfall coefficient of variation from the weather radar resolution to the high-resolution required for the analysis:

[Figure]

with the following caption: "**Figure S1.** Rainfall coefficient of variation (CV) power law surface-fit for temporal resolution of 5 min. X-axis refer to the domain area, Y-axis refer to the dimension of the rainfall grid cell and Z-axis represent the rainfall CV. Data was derived from MeteoSwiss weather radar system for domain areas of 64-1024 km$^2$ and grid sizes of 2-16 km for domains overlapping the studied catchment. For the case study, rainfall was generated over an area of 2.25 km$^2$ using grid size of 100 m, thus a CV of 1.5 was used".

A short explanation was added to the text for the choice of rainfall spatial correlation structure used: "In addition, no direct measurements are available to estimate the small-scale rainfall spatial correlation structure for this region. The spatial structure was estimated using data from three dense rain-gauge networks (Moszkowicz, 2000; Müller and Haberlandt, 2016; Peleg et al., 2013), recording rainfall over small spatial distances (i.e. in the scale of $10^1$–$10^2$ m) and temporal scales (i.e. 5–10 min)". A new figure (Fig. S2) was added to the supplementary material:

[Figure]

with the following caption: "**Figure S2.** Small-scale spatial correlation structure for 5-min temporal resolution as was recorded using dense rain-gauge networks in Germany, Poland and Israel. The average correlation structure of the three (expressed by the three-parameter exponential function given in the figure) was used in this study".

3. The temporal resolution adopted in the study (10 min) may be too coarse for urban applications and may result in smoothing of urban flows, which may in turn result in underestimation of flow extremes (as indicated in Ochoa-Rodriguez et al. (2015), temporal resolutions <= 5 min are required for urban hydrological applications, with resolutions of 10 min leading to large underestimation of peak flows. Likewise, Wang et al. (2015) showed results of flow simulations resulting from rainfall inputs at temporal resolutions from 1 to 10 minutes and compared them against flow observations; the results associated to 10 min rainfall inputs largely underestimated observed flow peaks and resulted in 'distorted' hydrographs). I understand that the temporal resolution of choice was likely constrained by the resolution at which rain gauge rainfall records were available. Please discuss the implications and limitations of the temporal resolution of choice and clearly mention in the future work section that tests should be conducted at finer temporal resolutions.

The rainfall temporal resolution was indeed set to 10 min following the temporal resolution of observed rainfall records. We do agree that temporal resolution of rainfall data used as model input significantly influences flow dynamics including peak flows. We added the following sentence in the manuscript in section 3.2: "For this analysis, rainfall was generated with a spatial resolution of 100 m x 100 m for a domain size of 1.5 km x 1.5 km (see mesh in Fig. 1) and a temporal resolution of 10 min. *For urban drainage applications 10 min can be considered a rather coarse temporal discretization, however we searched consistency with the observed rainfall record which is only available in 10 min resolution.*".

We are fully aware of that this is a suboptimal compromise. However, considering the fact that we calibrated the model using a one-year measured flow reference record and calibration results clearly show that the model can reproduce flow dynamics adequately, we think that using a 10 min rainfall resolution is more justified than generating rainfall data with a higher temporal resolution than the observed rainfall reference.

4. The current title of the paper is rather misleading and I would suggest changing it to better reflect the purpose and focus of the study. For example, the focus on extreme values / return period should somehow be mentioned. Furthermore, I would suggest changing 'temporal variability' to 'natural climatological variability'. The term temporal variability conveys the idea of temporal resolution, which, as described above, is not the purpose of this study and is in fact one of its shortcomings.

We agree with the reviewer comment. A new title for the manuscript is suggested: "Partitioning the impacts of spatial and climatological rainfall variability in urban drainage modelling".

Finally, we would like again to express our thanks to the Editor and to the two reviewers who have helped us to significantly improve the paper.

Sincerely,

Nadav Peleg and Frank Blumensaat on behalf of the authorship

[revised manuscript text omitted]

---

## Editor Decision (ED1)

Editor decision, after minor revision (Submitted 27 Jan 2017):

The authors have replied to all the reviewers' comments in their rebuttal and made changes to the manuscript accordingly. However, some of the comments merit a more substantial and critical discussion than is currently provided.

The authors conclude that climatological variability contributes more to flow variability than spatial rainfall variability. This makes sense for the relatively small catchment they investigated, as 30 years of rainfall are likely to represent more variability than rainfall over a domain of only 77 ha (or less, the subcatchments are max 30 ha). Still, the results on spatial rainfall variability obtained in this study are strongly dependent on the choices that have been made in the rainfall downscaling procedure (see also comment nr2 by reviewer 2), since the original spatial scale of the rainfall data is larger than the catchment scale (2x2km). For example, it is assumed that the wet area ratio is always equal to one during storms (and zero in between storms), which strongly limits the degree of spatial variability that can occur. In reality, even if the scale of a storm is typically larger than a few km2, more and more zero rainfall pixels would occur within the storm domain as one scales down to smaller scales. The impact of the assumptions made for the downscaling procedure on the partitioning results should be more critically discussed.

Another point that requires more discussion is how the uncertainty of the model used in this study might influence the results (see also comment reviewer 2 regarding use of 10 minute temporal resolution). The authors refer to Tokarczyk et al. (2015) for information about model calibration. In this paper, Nash-Sutcliffe efficiency values of 0.70-0.75 were reported and simulated versus observed hydrograph results show relative errors in peak flow estimates of over 25%, for flows up to about 300 l/s. The range of flows that is analysed in the partitioning study is far outside this range (1000-3000 l/s), while variability associated with rainfall climatology and spatial variability is of the same order of magnitude as the model uncertainty (~25%).

A final point the authors could discuss more critically is the fact that they use data over a period of 30 years and draw conclusions for return periods up to 30 years. In particular, they conclude that the contribution of spatial rainfall variability becomes more important for larger return periods (> 10 years), but statistics beyond 10 years are quite unreliable for a 30 year time series, given the small sample sizes in the extreme tail of the statistical distribution.

A few comments with respect to the figures the authors have provided in the supplement to the paper:
- Figures S6 and S7: these are very insightful figures presenting the main results of the partitioning study. It is not clear why the authors have chosen to show results for location B, where flow is controlled by a throttle, in the main body of the paper, while they present results for locations A and C in the supplement. I would suggest to present and discuss all 3 figures in the main body of the paper.
- Figure S1: this figure presents parameters used for the spatial downscaling procedure, but the interpretation of the figure isn't entirely clear. In the caption: "For the case study, rainfall was generated over an area of 2.25 km2 using grid size of 100 m, thus a CV of 1.5 was used." Yet in the figure, the domain area seems to span 64-1024km2 (doesn't go down to 2.25 km2) and the grid size 2-16km. Where do we read the CV value of 1.5 for domain size 2.25 km2 and grid size 100m from the figure?

---

## Author Response (AR2)

February 17, 2017

Dr Marie-Claire ten Veldhuis
Editor
Hydrology and Earth System Sciences (HESS)

RE: hess-2016-530

Dear Editor,

Please find enclosed the revised manuscript: "Partitioning the impacts of spatial and climatological rainfall variability in urban drainage modelling" by Nadav Peleg, Frank Blumensaat, Peter Molnar, Simone Fatichi and Paolo Burlando. The manuscript has been revised according to the comments of the Editor. We would like to thank the Editor for her efforts and constructive comments.

**Editor:**

General
The authors have replied to all the reviewers' comments in their rebuttal and made changes to the manuscript accordingly. However, some of the comments merit a more substantial and critical discussion than is currently provided.

Specific comments
1) The authors conclude that climatological variability contributes more to flow variability than spatial rainfall variability. This makes sense for the relatively small catchment they investigated, as 30 years of rainfall are likely to represent more variability than rainfall over a domain of only 77 ha (or less, the subcatchments are max 30 ha). Still, the results on spatial rainfall variability obtained in this study are strongly dependent on the choices that have been made in the rainfall downscaling procedure (see also comment nr2 by reviewer 2), since the original spatial scale of the rainfall data is larger than the catchment scale (2x2km). For example, it is assumed that the wet area ratio is always equal to one during storms (and zero in between storms), which strongly limits the degree of spatial variability that can occur. In reality, even if the scale of a storm is typically larger than a few km2, more and more zero rainfall pixels would occur within the storm domain as one scales down to smaller scales. The impact of the assumptions made for the downscaling procedure on the partitioning results should be more critically discussed.

We agree with the Editor statement that the results on spatial rainfall variability obtained in this study are dependent on the choices that have been made in the rainfall downscaling procedure. However, the suggestion of the Editor that we are underestimating the rainfall spatial variability in small urban catchments with a wet area ratio equal to one is generally not true, especially when high rainfall intensities are considered. This has been shown using a dense rain-gauge network for a different location but with a similar spatial characteristics in past studies of the authors (Peleg et al., 2013; 2016). There we have found that convective events (rainfall intensity >10 mm h$^{-1}$) were characterized with a mean wet area ratio of 0.88 for a much larger domain (4 km$^2$) than the domain that was analyzed in this study (2.25 km$^2$). It is important to realize that our downscaling methodology generates a spatial distribution of rain intensities which cover a large range, many pixels having extremely low intensity which would not even be detected by a typical rain-gauge. For example, 29% of the rainfall intensities simulated for Case 4 (chosen as an example) were lower than 0.1 mm h$^{-1}$ (and 39% were <0.2 mm h$^{-1}$), which can be considered as insignificant, practically zero, rainfall. As a result we do not think that the degree of spatial variability in our downscaling is limited (underestimated) due to the fact that the wet area ratio was fixed.

We also agree with the Editor that the assumptions made for the downscaling procedure are to be discussed, and in fact most of the relevant assumptions made for the downscaling were mentioned in the discussion part at the manuscript. The paragraph dealing with the rainfall generator in the discussion section was modified to include additional aspects that were not discussed before: "The rainfall generator was used to simulate rainfall for the weather radar subpixel scale, i.e. in a finer spatial resolution than can be estimated using the MeteoSwiss radar. The rainfall

data required for a complete validation of the rainfall generator for this resolution can be obtained from a dense rain–gauge network (for networks examples see Muthusamy et al., 2016; Peleg et al., 2013) but such a network is not available in the analyzed region. **Four aspects** are discussed in the light of missing information for the subpixel scale **(i.e. rainfall downscaling process)**: (i) the rain fields are simulated following a lognormal distribution. We assume that the non-zero part of the subpixel spatial rainfall distribution follows the observed lognormal distribution that is recorded by the weather radar for this region (as in Paschalis et al., 2014; Peleg et al., 2016). A different spatial rainfall distribution will significantly affect the results of the extreme rainfall; (ii) we assume that occurrence and intensity statistics are equal for each of the grid cells, i.e. no spatial correlation is applied for the rainfall occurrence or intensity. This means that orography, distance from the lake, and urban micro-climate effects are not considered; **(iii) we assume that the rainfall spatial correlation structure for this region follows the average structure obtained from estimates made in dense rain–gauge networks in Poland, Germany and Israel (Moszkowicz, 2000; Müller and Haberlandt, 2016; Peleg et al., 2013). The exact impact of the spatial correlation structure at the radar subpixel scale in urban drainage studies is yet to be determined; and (iv) we assume that the power-law used for the scaling of the rainfall coefficient of variation is continuous from the weather radar to its subpixel scale, and it is not affected by a scale-break. Overestimation of the rainfall coefficient of variation will affect the rainfall spatial variability and therefore impact the partitioning results**".

2) Another point that requires more discussion is how the uncertainty of the model used in this study might influence the results (see also comment reviewer 2 regarding use of 10 minute temporal resolution). The authors refer to Tokarczyk et al. (2015) for information about model calibration. In this paper, Nash-Sutcliffe efficiency values of 0.70-0.75 were reported and simulated versus observed hydrograph results show relative errors in peak flow estimates of over 25%, for flows up to about 300 l/s. The range of flows that is analysed in the partitioning study is far outside this range (1000-3000 l/s), while variability associated with rainfall climatology and spatial variability is of the same order of magnitude as the model uncertainty (~25%).

We agree with the Editors' opinion that limitations due to the comparatively coarse temporal rainfall resolution (10 min) can be discussed more in depth. We furthermore discuss the relevance of the overall model accuracy (relative error for peak flows of 25%, NSE ~0.7) in the light of the evaluated rainfall variability:

1a) We agree that using a 10 min rainfall input represents a critical point, in particular when the average response time in the considered catchment is in the order of minutes (see Tokarczyk et al., 2015; Fig. 8). Still, we achieve a reasonable hydraulic model performance when validating the model results against flow observations at the catchment outlet (Location B in the present manuscript – the same, but not labelled, in Tokarczyk et al., 2015). As shown in Appendix A (at the end of this letter), the original calibration was focused not just on peak-flow performance, but also on the performance of time-to-peak and runoff balance. With this we follow recommendations given in Krause et al. (2005). The calibration for flows at location B yielded in NSEs >0.8 for individual events; overall performance for longer periods was estimated with NSE ~0.7 (see Appendix A and Tokarczyk et al., 2015). We consider the calibration procedure as adequate and accept the model performance as reasonable.

1b) To further justify the use of 10 min rainfall data, we carried out additional benchmark simulations with partial data from a rain—gauge recording rainfall at finer temporal resolution (1 min). In Appendix A, we show that both rainfall records, being independently measured and of different temporal resolution, are similarly adequate to reproduce flow dynamics in the considered catchment. However, the rather vague meta-information on the 1 min rainfall record (measurement accuracy, gauge type, mode of operation) does not suggest the use of this fine temporal resolution data in a scientific publication. We are fully aware of the relevance of an adequate temporal rainfall resolution for urban drainage modelling studies, but here we do not want to dilute the focus of the study (spatial and climatological variability) by including a controversial issue, which ultimately cannot be clarified on the basis of available data. Overall, we believe that based on the results of this benchmark and the more-than-average model validation (1-year field observations as reference; not just one event; various performance criteria) we show that flow dynamics can be adequately reproduced, even though we feed the model with a 10 min rainfall input.

2) The fact that the NSE for flows observed/simulated at Location B is 0.7-0.75 for the validation period, and that peak flows at the same location were deviate up to 25% can be attributed to several different sources, e.g. model structure uncertainty, inadequate model calibration, measurement error in the flow reference, and model input data uncertainty.

We further agree that the range of peak flows that were analyzed in this study varies within two orders of magnitude (300–3000 ls$^{-1}$) whereas the model validation was only conducted for location B, at which peak flows hardly exceed 300 ls$^{-1}$. At this particular location, modelled peak flows have been assessed, i.e. deviate up to 25% (positive and negative) compared from observed flows. Locations A and C show higher peak flows, but the model performance at these locations can only indirectly be evaluated through location B. Assuming similarly high deviations (~25%) at locations A and C would be a first assumption, but it cannot directly be verified. However, considering the fact that the same hydrodynamic model (inherent with the same deficiency) has been used for all simulations, we assume that the error due to the model structure and calibration equally influences each single simulation. Since the focus of the study had not been at the quantification of absolute peak flows for this particular catchment in Lucerne, but on a relative contribution of different rainfall variabilities to flow variations, any other realistic urban catchment could have been used. We chose the Lucerne catchment as real-life case because it is representative and it had been thoroughly researched and verified, including the quantification of an overall modelling error based on measured observations. We will make this model error, quantified for location B, transparent in the revised version of the manuscript, but we are confident that the conclusions regarding spatial and climatological rainfall variability do not become invaluable because peak flow deviations are in the same order of magnitude as researched rainfall variabilities.

Consequently, a new paragraph was added in the "results and discussion" section, summarizing the above points and main massage from Appendix A below: "**Rainfall records were obtained from a rain–gauge that is located about 2 km west of the case study catchment. It was chosen for three main reasons (i) its proximity to the catchment; (ii) it has a sufficiently long record (34-year) that is adequate for statistical climatology analysis; and (iii) records have been verified by MeteoSwiss ensuring sufficient consistency. In contrast to these advantages, the 10 min temporal resolution of the rain data requires critical consideration when simulating the dynamics of the flow response (e.g. Ochoa-Rodriguez et al., 2015), particularly as the average flow response in the investigated catchment is in the order of minutes. However, we achieve a reasonable hydraulic model performance when validating the model against flow observations at the catchment outlet (location B), considering peak flow, time-to-peak and flow balance (see Tokarczyk et al., 2015). Low flow volume errors (±5%) and Nash-Sutcliffe-Efficiencies of >0.8 for individual events, i.e. >0.7 for longer periods, support the fact that the flow dynamics are reproduced adequately. Remaining peak flow errors of up to 25% reflect existing deficiencies stemming from multiple sources, e.g. inadequate model structure, insufficient model calibration, measurement errors in flow reference data and model input data uncertainty. Considering that the same hydrodynamic model has been used for all the simulations, it is likely that the error due to model structure and calibration do not introduce a consistent bias to the variability partitioning. A complete investigation of the model hydrodynamic uncertainties will provide additional insights but it will be difficult to constrain with the current length of available flow data**".

3) A final point the authors could discuss more critically is the fact that they use data over a period of 30 years and draw conclusions for return periods up to 30 years. In particular, they conclude that the contribution of spatial rainfall variability becomes more important for larger return periods (> 10 years), but statistics beyond 10 years are quite unreliable for a 30 year time series, given the small sample sizes in the extreme tail of the statistical distribution.

We do agree that rainfall uncertainties become important when the record length is similar, or smaller, than the estimated return period. Marra et al. (2016) recently demonstrated how uncertainties increase with return period and with smaller record length for different climate regions and durations (a snapshot taken from Figure 9 in Marra et al. is given at the right, illustrating this point). We acknowledge the need to notify the readers of this point, stating that our results represent the climate uncertainties derived from a 30-year data period. This point in fact strengthens the results of this study, as with a larger sample period the climate uncertainties for a longer return period are expected to decrease, implying that the contribution of the spatial rainfall variability for longer return period may be even higher than reported in this study.

[Figure]

The following changes to the text were made in the discussion section: "While the use of spatially distributed rainfall data can supply valuable information for sewer network design (based on rainfall with return periods from 5 to 15 years), it will become even more important when performing flood risk assessments of

extreme events (larger return periods). **A 30-year record was used in this study, which can be regarded as the minimum period for IDF/FDF analysis. Since uncertainties in climate statistics decrease with a longer observational record (e.g. Marra et al., 2016), the contribution of the additional spatial variability for larger return periods might be even greater than presented here. However, a longer period of observation is required to confirm this assertion**".

A few comments with respect to the figures the authors have provided in the supplement to the paper:

4) Figures S6 and S7: these are very insightful figures presenting the main results of the partitioning study. It is not clear why the authors have chosen to show results for location B, where flow is controlled by a throttle, in the main body of the paper, while they present results for locations A and C in the supplement. I would suggest to present and discuss all 3 figures in the main body of the paper.

Location B has initially been selected to be shown in the main part of the manuscript, because the model had been validated against field data at this particular point. Figures S6 and S7 were moved to the main manuscript, as suggested.

5) Figure S1: this figure presents parameters used for the spatial downscaling procedure, but the interpretation of the figure isn't entirely clear. In the caption: "For the case study, rainfall was generated over an area of 2.25 km2 using grid size of 100 m, thus a CV of 1.5 was used." Yet in the figure, the domain area seems to span 64-1024km2 (doesn't go down to 2.25 km2) and the grid size 2-16km. Where do we read the CV value of 1.5 for domain size 2.25 km2 and grid size 100m from the figure?

The surface in Figure S1 presents the rainfall coefficient of variation that was estimated using MeteoSwiss weather radar system (i.e. for domain areas of 64-1024 $km^2$ and grid cell length of 2-16 km). The surface was computed using a two-parameter power function with $R^2$ of 0.99. The rainfall CV used in this study is an extrapolation of the surface, calculated using the two-parameter function, as the domain size and grid cell length are much finer than the observed data. To better explain this in the text, the figure caption was modified as follows: "**A surface representing the** rainfall coefficient of variation (CV) for temporal resolution of 5 min **as analyzed using MeteoSwiss weather radar system over the study area**. X-axis refer to the domain area (64–1024 $km^2$), Y-axis refer to the dimension of the rainfall grid cell (2–16 km) and Z-axis represent the rainfall CV. **The surface was generated using a two-parameter power function that was fitted to the data [$Z(X,Y)=0.54X^{0.18}-51.57Y^{0.007}+51.63$] with a coefficient of determination of 0.99. For the case study, rainfall CV was extrapolated from the observed surface using the above function as follows: X=2.25 $km^2$ (area of the case study for which rainfall was simulated), Y=100 m (simulate rainfall grid cell length) and Z= 1.5 (simulated rainfall CV)**".

We would like again to express our thanks to the Editor and we look forward to hearing from you regarding our submission. We would be glad to respond to any further questions and comments that you may have.

Sincerely,

Nadav Peleg, Frank Blumensaat and Peter Molnar on behalf of the authorship

**Appendix A - Influence of temporal rainfall resolution on hydraulic simulations**

Motivation

To research the influence of the temporal resolution of rainfall input data on the representation of flow dynamics we compare results of hydrodynamic simulations with different rainfall inputs (1 and 10 min temporal resolution).

Rainfall data

Case study simulations were carried out with a rainfall record from the MeteoSwiss rain—gauge station *'Allmend'* (10 min temporal resolution). The same series had been used for research outlined in Tokarczyk et al. (2015). This rainfall record had been chosen for several reasons: (i) the gauge is in close proximity to the catchment (2 km); (ii) rainfall records are sufficiently long (more than 30-year) and are consistent; and (iii) rainfall data were thoroughly verified by MeteoSwiss and by the authors. Seen in the light of these advantages, we accepted that the available temporal resolution of this data is only 10 min. By doing so we also assumed that the 10 min rainfall input for the hydrodynamic model would be adequate to capture short-term flow dynamics observed in the case study catchment.

This assumption is justified through a thorough model calibration/validation procedure (split-sample approach) described by Tokarczyk et al. (2015). For more than 40 different rainfall events in 2014/2015, model performance has been validated for the criteria peak flow, time-to-peak and runoff balance. Performance has been rated adequate since flow volume balance (Bias) error was consistently below 5% and Nash-Sutcliffe-Efficiency (NSE) for individual events was always higher than 0.7 (often higher than 0.8); for longer periods somewhat lower due to inclusion of dry weather periods.

In addition to this, we carried out benchmarking simulations using the 10 min rainfall and rainfall records from two nearby stations (*'Eichhof', 'Matthof'*) in 1 min resolution. This data stems from a less systematically organized regional rain—gauge network (manual digitalization; no automatic data transfer; no central data management; see rain—gauge locations in Fig. A1). Furthermore, this record only contains data for selected periods (May - June 2015), partly in fragments (Fig. A2). Based on the *comparing analysis of rainfall* illustrated in Fig. A2 the following conclusions can be drawn:

1. 1 min and 10 min data show similar dynamics, whereas for the considered period (01-May-2015 until 25-June-2015) the *'Allmend'* station records show lower rainfall heights than the *'Matthof'* and *'Eichhof'* records (Fig. A2). For the abovementioned period the full deviation between *'Eichhof'* and *'Allmend'* data is 5.7%. It can be considered small, i.e. within the range of a typical rainfall measurement error.

2. For further analysis (to what extend temporal resolution will influence hydraulic modelling results) we only use records from *'Eichhof'* (1 min) and *'Allmend'* (10 min); *'Matthof'* (1 min) data are excluded due to missing data.

3. Events which are considered in detail are: (A) 01-May-2015 – 06-May-2015 and (B) 14-Jun-2015 – 25-Jun-2015.

Hydraulic modelling benchmark using the calibrated SWMM model

Figures A3—A6 illustrate the model performance for hydraulic simulations driven by two different rainfall inputs of different temporal resolution (MeteoSwiss station *'Allmend'*: 10 min; *'Eichhof'* station: 1 min). Similar values for the performance measures *NSE* and *Bias* suggest that in this particular case, temporal rainfall input resolution does not significantly influence the hydraulic model performance (see Fig. A3-A6 for measured hydrographs at location B; Fig. A7-A8 simulated flow series are compared among each other for two different locations A and B). This is, indeed, contrary to what had been expected (and reported elsewhere). Numerous previous studies underline the relevance of an adequate temporal resolution for urban hydrology applications (typically <= 5 min and in accordance with average response time of the catchment) but the expected deviations for different rainfall inputs could not be observed in the present case.

[Figure]

Figure A1: Locations of different rain—gauges used in this analysis

[Figure]

Figure A2: cross-comparison between 1 min (Eichhof, Matthof) and 10 min (Allmend – MeteoSwiss station) rainfall data. Upper chart shows full period for which 1 min data are available. Lower charts show events A and B.

[Figure]

Figure A3: simulated vs. observed hydrographs at location B for a selected period at which rainfall data from different gauges were available. Here rain input is taken from MeteoSwiss rain-gauge 'Allmend' (10 min).

Figure A4: simulated vs. observed hydrographs at location B for a selected period at which rainfall data from different gauges were available. Here rain input is taken from rain gauge 'Eichhof' (1 min).

[Figure]

Figure A5: simulated vs. observed hydrographs at location B for a selected period at which rainfall data from different gauges were available. Here rain input is taken from MeteoSwiss rain-gauge 'Allmend' (10 min).

Figure A6: simulated vs. observed hydrographs at location B for a selected period at which rainfall data from different gauges were available. Here rain input is taken from rain gauge 'Eichhof' (1 min).

[Figure]

Figure A7: cross-comparison of two *simulated* hydrographs at location B. Simulations are driven by rainfall data from different sources and with a different temporal resolution (Allmend - MeteoSwiss: 10 min; Eichhof: 1 min).

[Figure]

Figure A8: cross-comparison of pipe flow at the inner catchment node, i.e. location A. Simulations are driven by rainfall data from different sources and with a different temporal resolution (10 min: Allmend - MeteoSwiss; 1 min: Eichhof). Deviations is more obvious than for constraint flow at the catchment outlet (location B – see Figure A7).

Limitations

Additionally acquired 1 min rainfall data must be considered with care for two reasons: (i) little information is available on monitoring devices, i.e. measurement principle, mode of operation and maintenance intervals; and (ii) despite the fact we received the data from a trustworthy source (retired urban drainage engineer who maintains rain—gauges voluntarily), we were not able to fully verify this data. Measurement accuracy remains difficult to assess. We compared the daily sum of the two gauges with the data from the MeteoSwiss station *'Allmend'* with very little deviation, but doubts remain regarding the reliability of the 1 min records, particularly for the sub-10 min scale.

To Conclude

Analyses explained above lead to the conclusion that the 10 min rainfall data from the MeteoSwiss station *'Allmend'* used for the analysis in the paper can be considered as representative, despite the fact that flow time in the catchment is in the same order of magnitude as temporal resolution of rainfall data used.

The comparison of simulations run with different rainfall input shows a high degree of agreement despite the temporal resolution of rainfall input data differs by factor 10 (10 min – 1 min). Flow dynamics at location B (location at which flow reference data had been available) are sufficiently well reproduced by either of the rainfall inputs.

However, final doubts remain regarding the measurement accuracy of rainfall records from the 'Eichhof' station on sub-10min scale. This is the reason why we prefer not to include this additional data source in the original manuscript, nor in the supplementary material.

[revised manuscript text omitted]